# How does RL Post-training induce skill composition? A Case Study on Countdown

## Abstract

While reinforcement learning (RL) successfully enhances reasoning in large language models, its role in fostering compositional generalization (the ability to synthesize novel skills from known components) is often conflated with mere length generalization. To this end, we study what RL post-training teaches about skill composition and how the structure of the composition affects the skill transfer. We focus on the Countdown task (given $n$ numbers and a target, form an expression that evaluates to the target) and analyze model solutions as expression trees, where each subtree corresponds to a reusable subtask and thus can be viewed as a "skill." Tracking tree shapes and their success rates over training, we find: (i) out-of-distribution (OOD) generalization to larger $n$ and to unseen tree shapes, indicating compositional reuse of subtasks; (ii) a structure-dependent hierarchy of learnability—models master shallow balanced trees (workload is balanced between subtasks) before deep unbalanced ones, with persistent fragility on right-heavy structures (even when the composition depth is the same as some left-heavy structures). Our diagnostic reveals what is learned, in what order, and where generalization fails, clarifying how RL-only post-training induces OOD generalization beyond what standard metrics such as pass@k reveal.

## 1 Introduction

A key objective in training large language models (LLMs) is to develop robust reasoning abilities that generalize to novel problem instances. Reinforcement learning (RL) is a widely adopted technique for this purpose (Ouyang et al., 2022; Grattafiori et al., 2024; OpenAI et al., 2024; DeepSeek-AI et al., 2025), yet the precise mechanism by which it improves reasoning remains poorly understood. The prevailing hypotheses suggest that RL either sharpens the model's probability distribution over known reasoning paths (Huang et al., 2025; Yue et al., 2025) or encourages in-context exploration via chaining in longer solution sequences (DeepSeek-AI et al., 2025; Setlur et al., 2025). However, these notions overlook a crucial aspect of intelligence: the ability to compose known skills in novel ways.

Current evaluations often conflate two fundamentally different modes of generalization. The first, *length generalization*, is the ability to solve problems of greater sequential depth by iterating a known algorithmic structure. The second, *compositional generalization*, is the ability to solve problems of a fixed depth by synthesizing a novel algorithmic structure from familiar components. While complex reasoning often involves both, we argue that a rigorous understanding requires operationalizing and testing these two modes of generalization independently. Without a clear way to distinguish them, it is difficult to ascertain the true algorithmic contribution of fine-tuning methods like RL.

To formalize this distinction and enable empirical investigation, a controlled setting is required. We utilize the Countdown[1] task, which serves as a clean testbed where the structure of a solution is isomorphic to a binary expression tree. This allows for an unambiguous, canonical representation of the reasoning pattern used to solve a problem.

This pattern framework allows us to move beyond a monolithic view of the solution to a nuanced skill-based view. Solving a complex pattern like $A \times (B - C/D)$ requires a composition of distinct

---

[1]Countdown is a task where given $n$ integers and a target, the model needs to output an expression using each integer once and operators $\{+, -, \times, \div\}$ that attains the target. For example, given $[4, 3, 2, 2]$ and a target 16, one solution would be $2 \times 2 \times 3 + 4 = 16$.

abilities: *decomposition skills* to identify how to explore and correctly partition the solution (e.g., `factor_identification`: identifying the factors of the target to explore potential candidates) and the recursive *sub-problem skills* to solve the resulting sub-patterns (e.g., generating an expression for $B - C/D$ given the remaining three numbers and a modified target).[2] The tree structure of the pattern determines which atomic skills are combined, in what order, and how they are grouped.

By tracking the development of complex skills for each pattern during training, we probe two fundamental questions: 1) To what extent does RL equip models with the ability for genuine structural composition? and 2) How does the underlying compositional structure of a problem affect its learnability?

We summarize our main contributions:

- **Pattern framework.** We introduce a framework for disentangling length and compositional generalization (Section 2).
- **Models generalize to larger puzzle sizes.** Base models (1B-7B) trained on patterns from $n = 3, 4$ generalize to $n = 5, 6$, where skills for smaller patterns need to be recombined to solve the puzzles (Section 4.1).
- **Pattern structure determines difficulty.** Within the same input length $n$, we observe a clear hierarchy of learnability based on the structure of the pattern: shallow balanced structures (with equal amount of "workload" across subtasks) are easier than deep unbalanced structures (Section 4.2). Within the same depth, right-heavy trees, which require committing to a plan ahead of the complex subroutine appearing later, are particularly hard (Section 4.3).
- **Models generalize to unseen patterns, signaling compositional reuse.** When a particular $n = 3$ pattern was removed from training, 1.5B model can reconstruct the pattern and its higher-order extensions (Section 4.4).

Overall, our findings show that RL post-training imparts length generalization and partial compositional generalization, with difficulty governed primarily by compositional structure. This result motivates targeted curriculum design as a promising direction for future work. We hypothesize that during RL post-training, once early generalization gains appear, introducing a minimal set of structurally hard, near-frontier examples could offer a more sample-efficient method for pushing the model's reasoning capabilities.

## 2 METHODOLOGY: A STRUCTURAL FRAMEWORK TO DISENTANGLE LENGTH AND COMPOSITIONAL GENERALIZATION

Standard metrics for reasoning, such as *pass@k*, *response length*, or *number of attempts* offer an incomplete picture of a model's capabilities. They fail to distinguish between a model's ability to iterate a known procedure and the more formidable challenge of synthesizing a novel one. To address this, we introduce a formal framework that disentangles these phenomena by analyzing the underlying computational structure of arithmetic solutions, which we apply to the COUNTDOWN task.

### 2.1 CANONICAL PATTERNS OF REASONING

We define an *atomic skill* as the application of a single binary operator from $\{+, -, \times, \div\}$. Then the ability to solve a COUNTDOWN puzzle with an arithmetic expression requires a *skill composition*. By replacing numbers with placeholder symbols, an arithmetic expression can be abstracted (e.g., $9/3 + 1 \rightarrow$ `A/B+C`) and parsed into a binary tree, which provides insights into how the atomic skills are combined.

To ensure that syntactically different but algorithmically identical expressions (e.g., $(A + B) + C$ and $C + (A + B)$ are treated as a single computational procedure, we apply a principled normalization process that maps them to a unique canonical *pattern* (Figure 1). This process, detailed in Appendix A.1, resolves ambiguities arising from properties like commutativity and associativity, allowing us to analyze the underlying computational graph and distinguish a simple iterative structure like $A + B + C + D$ from a complex, nested one like $A/(B - C/D)$.

---

[2]The atomic skills would be mastering the four operators: $a + b$, $a - b$, $a \times b$, and $a \div b$.

We classify the structure of a computational pattern by its shape signature $[x] \circ [y]$, denoting the root operator $\circ$ and the number of leaf operands $x$ and $y$ in its left and right sub-trees. This partitioning into left-heavy ($x > y$), balanced ($x \approx y$), and right-heavy ($x < y$) structures correlates with synthesis difficulty (see Figure 6). For instance, to generate a right-heavy pattern like $A/(B - C/D)$ (signature $[1] \div [3]$), a model cannot greedily select the root operator; it must first look ahead to the complex sub-expression $(B - C/D)$ and recognize its utility within the broader computation.

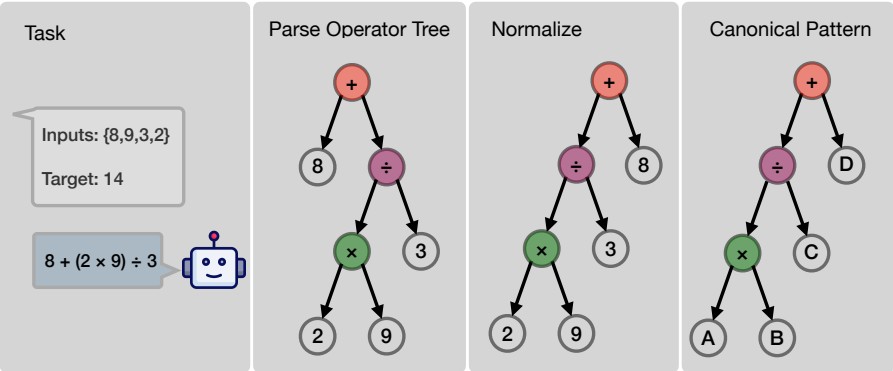

Figure 1: **From generated expression to canonical *pattern*.** The model-generated expression $8 + (2 \times 9) \div 3$ is parsed into an expression tree, normalized with respect to algebraic identities, and mapped to the unique canonical pattern $A \times B \div C + D$. Patterns are then grouped by their computational shape. Here, the root operator is "$+$", with a left sub-expression over three leaves $(A, B, C)$ and a right sub-expression over one leaf $(D)$. The signature $[3] + [1]$ serves as a compact representation of the pattern's structure at the root-level.

## 2.2 DISENTANGLING MODES OF GENERALIZATION

A primary contribution of our framework is its ability to formally disentangle two distinct and often-conflated axes of skill composition. We define **length generalization** as the ability to generalize to larger problem sizes than those seen during training, which typically requires using more atomic skills. We define **compositional generalization** as the ability to synthesize novel combinations of known atomic skills at problem sizes that remain within the training range.

For example, in COUNTDOWN, training on one puzzle size $n$ and evaluating on a larger puzzle size $N > n$ can measure length generalization, whereas training on a subset of patterns in a puzzle size $n$ and evaluating on some heldout set of patterns in the same puzzle size $n$ can measure compositional generalization.

Our framework also allows a deeper analysis into the effect of compositional structure on generalization. While some works (Yuan et al., 2025) may treat composition simply as a sequential application ("chaining") of atomic skills where problem size always coincides with compositional depth, skill compositions can happen in complex structures, where problem size alone cannot capture the difficulty level.

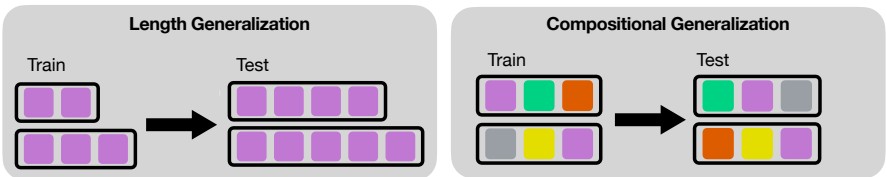

Figure 2: **Illustration of length and compositional generalization.** While length and compositional generalization are not mutually exclusive, they measure different abilities. A study of length generalization focuses solely on the number of skills, whereas a study of compositional generalization requires an analysis of the compositional structure.

### 2.3 Controlling for Distributional Bias in Dataset Generation

A rigorous analysis of structural difficulty requires that the distribution of problem types be controlled. Standard approach for creating COUNTDOWN datasets (Pan et al., 2025b; Gandhi et al., 2024; Stojanovski et al., 2025) proceeds as follows: 1) choose a random binary tree; 2) choose random operators, numbers; 3) then filter for valid expressions that evaluate to an integer in $[1, 99]$. This introduces severe distributional biases, creating a confounding variable between a pattern's frequency and its intrinsic complexity. There are two problems with this design:

- **Structural bias.** $+$ and $\times$ are commutative and associative, and many expressions involving the two operators may have the same underlying pattern and have a higher probability of being generated. For instance, the same $A + B + C + D$ puzzle can be generated in $4! \times C_3 = 120$ ways (e.g., $((1+2)+3)+4$ and $1+(4+(3+2))$), whereas an asymmetric pattern like $A/(B - C/D)$ only has one expression.

- **Selection bias.** The second, more severe bias stems from the common practice of filtering for solutions that evaluate to an integer within a fixed range, such as $[1, 99]$. When random inputs are drawn from the set range, $+$ or $-$ yields a result in this range approximately 50% of the time, whereas $\times$ or $\div$ do so only about 5% of the time. Consequently, each $\times$ or $\div$ is approximately $10\times$ more unlikely to appear in an expression.

These two biases together produce a skewed dataset where aggregate performance metrics are dominated by a model's ability on simple, additive patterns. To eliminate this confounder, we designed a different generation protocol. We generate examples by first selecting a target canonical pattern and then sampling numbers that satisfy it. This enforces a near-uniform distribution over all patterns, a methodological prerequisite for isolating and analyzing the true effect of compositional structure on reasoning difficulty. See Appendix A.2 for more details.

## 3 Experimental Setup

### 3.1 Training

We finetune `Qwen-2.5` models (1.5B, 3B, and 7B) (Qwen et al., 2025) with GRPO (Shao et al., 2024) on problems of size $n \in \{3, 4\}$. Each rollout receives a reward of 0.1 for correctly formatting the final expression and an additional reward of 0.9 if the expression is correct and valid.[3] We train for one epoch across three seeds, saving checkpoints every 50 gradient updates and discarding any unstable runs (see Appendix C.1). Full training details and results on other model families are available in Appendices A.2.3, A.3 and D.

### 3.2 Evaluation

We evaluate each checkpoint by sampling $k = 32$ outputs per held-out question (temperature $= 0.6$), with the maximum token length increasing for larger problem sizes (1024 for $n \in \{3, 4\}$, 2048 for $n = 5$, and 4096 for $n = 6$). We then compute the following metrics on the extracted final answers:

- **Pass@32 / All-correct@32:** The fraction of questions where at least one (Pass@32), or all (All-correct@32), of the $k$ sampled answers is correct

- **Average Accuracy:** The total number of correct answers across all questions and attempts, divided by the total number of answers generated (i.e., number of questions $\times k$).

- **Pattern Coverage:** The fraction of all possible canonical patterns that the model produces at least once across the evaluation set (`present`).

- **Per-Pattern Precision:** For any single pattern, this is the total number of correct solutions using that pattern, divided by the total number of times the model produced that pattern as a final answer across the entire evaluation set.

---

[3]An expression is *valid* if each of the $n$ provided numbers is used exactly once and operators are restricted to $\{+, -, \times, \div\}$. An expression is *correct* if it evaluates to the target number.

- **High-Precision Coverage:** The fraction of all possible canonical patterns for which the model's *per-pattern precision* (defined above) is at least $80\%$ (`reliable`).[4]

## 4 MAIN RESULTS

We first show that models achieve length generalization (Section 4.1), then demonstrate that this success is fundamentally governed by the compositional structure of the task (Section 4.2). We identify a "lookahead bottleneck" as the primary barrier to skill composition and provide definitive evidence that RL enables the synthesis of entirely novel reasoning patterns (Section 4.4). Finally, we isolate the effect of RL training by comparing against a supervised finetuning baseline (Section 4.5).

### 4.1 MODELS DEMONSTRATE LENGTH GENERALIZATION

RL finetuning effectively teaches models to apply learned reasoning procedures to problems of greater lengths than seen during training. Models trained only on $n \in \{3, 4\}$ problems show substantial accuracy improvements on held-out $n = 5$ puzzles (Figure 3) over the base model, with final Pass@32 reaching nearly 70% for the 1.5B model (Table 1). This generalization extends to $n = 6$, where models achieve Pass@32 around 40%. These results confirm that models successfully achieve *length generalization*.

Table 1: **Performance across model size.** Averaged across all successful seeds [5](**final** checkpoints for 1.5B/3B and **best** checkpoints for 7B). [6]All-correct@32 (All), Average Accuracy (Avg), and Pass@32 (Pass) on $n=3, 4$ (trained) and $n=5, 6$ (held out). Models learn $n = 3, 4$ patterns almost perfectly and generalize to larger puzzle sizes $n = 5, 6$, but the gap between Pass@32 and Average Accuracy increases with $n$, signaling that the generalization is not perfect.

| Model | $n=3$ | | | $n=4$ | | | $n=5$ | | | $n=6$ | | |
|---|---|---|---|---|---|---|---|---|---|---|---|---|
| | All | Avg | Pass | All | Avg | Pass | All | Avg | Pass | All | Avg | Pass |
| 1.5B | 0.98 | 1.00 | 1.00 | 0.63 | 0.83 | 0.94 | 0.13 | 0.36 | 0.68 | 0.00 | 0.09 | 0.36 |
| 3B | 0.92 | 0.99 | 1.00 | 0.59 | 0.83 | 0.95 | 0.12 | 0.39 | 0.68 | 0.02 | 0.15 | 0.40 |
| 7B | 0.99 | 1.00 | 1.00 | 0.66 | 0.88 | 0.98 | 0.15 | 0.46 | 0.75 | 0.02 | 0.15 | 0.39 |

Table 2: **Per-pattern precision averaged by pattern structure on** $n=4$**.** Averaged across all successful seeds (**final** checkpoints for 1.5B/3B and **best** checkpoints for 7B). Averaged by (a) root operator; (b) pattern with root operator $-$; (c) pattern with root operator $\div$. Models struggle more on $-, \div$ as the root operator. Shapes in the form of $[1] - [3]$ or $[1] \div [3]$ are particularly challenging. Within each column, the highest number is **boldfaced** and the second highest number is underlined.

| (a) by root operator | | | | (b) by subtree shape ($-$) | | | | (c) by subtree shape ($\div$) | | | |
|---|---|---|---|---|---|---|---|---|---|---|---|
| **Operator** | **1.5B** | **3B** | **7B** | **Shape** | **1.5B** | **3B** | **7B** | **Shape** | **1.5B** | **3B** | **7B** |
| $+$ | 0.86 | 0.85 | **0.91** | $3-1$ | 0.81 | 0.78 | 0.84 | $3 \div 1$ | 0.77 | 0.77 | 0.83 |
| $-$ | 0.83 | 0.80 | 0.86 | $2-2$ | **0.92** | **0.89** | **0.90** | $2 \div 2$ | **0.88** | **0.85** | **0.91** |
| $\times$ | **0.90** | **0.86** | **0.91** | $1-3$ | 0.73 | 0.76 | 0.86 | $1 \div 3$ | 0.61 | 0.76 | 0.80 |
| $\div$ | 0.75 | 0.78 | 0.83 | | | | | | | | |

---

[4]The 80% threshold checks if the model can execute a pattern with a reasonable reliability, but allows some mistakes. The main findings are robust to reasonable variations of the threshold (70-90%). See Appendix G.

[5]2 seeds for 1.5B/3B and 3 for 7B. Due to compute budget, we evaluate only 1 seed per model for $n = 6$. See Appendix B.1 for the version with the standard deviation and the comparison against the base model.

[6]7B models are over-trained before the end of 1 epoch of training. We select the checkpoint right before the training reward declines. See Appendix B.1 for more details.

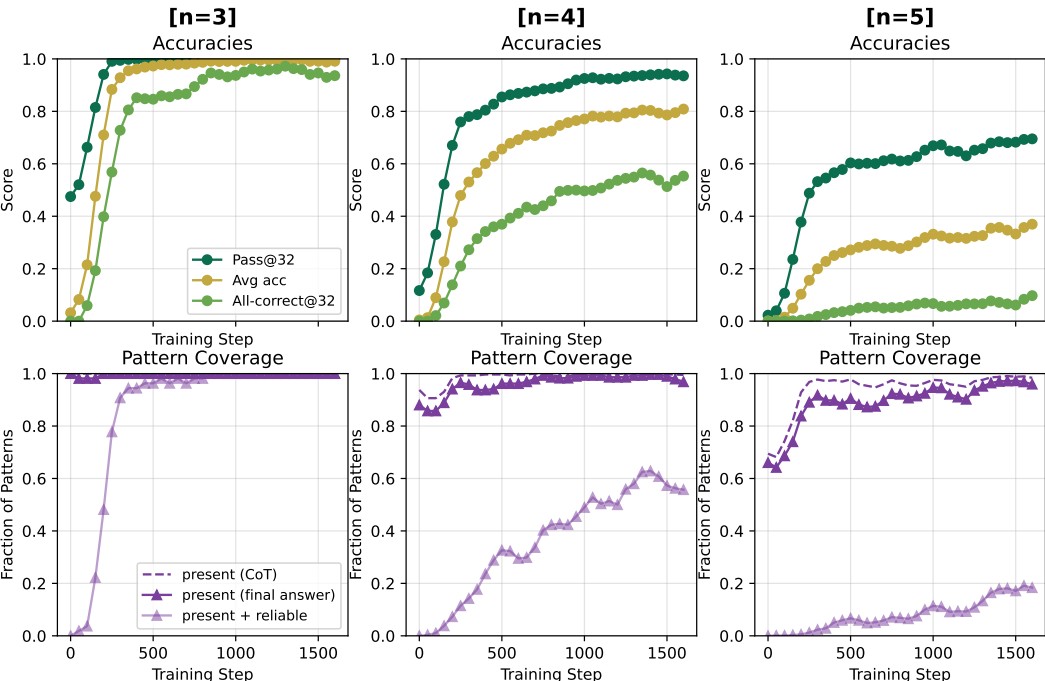

Figure 3: **Successful length generalization, but a gap between pattern discovery and reliable execution.** We train Qwen2.5-1.5B on problem sizes $n \in \{3, 4\}$ and evaluate on held-out data for $n \in \{3, 4, 5\}$. **(Top)** The model demonstrates strong *length generalization*: accuracy on the held-out $n = 5$ problems improves over the base model (leftmost data point). **(Bottom)** The model also identifies almost all compositional patterns for $n = 5$, but the ability to reliably execute the correct pattern lags significantly behind its discovery. This indicates that the primary challenge for generalization is procedural reliability, not abstract pattern identification.

## 4.2 COMPOSITIONAL STRUCTURE DETERMINES LEARNABILITY

However, the improvement in an aggregate accuracy masks the true determinant of difficulty. The ability to generalize in length is fundamentally constrained by the compositional structure of the pattern. This is evident in the out-of-distribution $n = 5$ evaluation in Figure 4, which reveals the same hierarchy of difficulty seen on the training distribution (with $n = 4$): balanced patterns are learned most readily, while right-heavy patterns remain the most challenging. This demonstrates that compositional complexity, not problem length, is the primary predictor of reasoning difficulty, even when generalizing to unseen lengths.

## 4.3 THE "LOOKAHEAD BOTTLENECK" IS A FUNDAMENTAL BARRIER TO SKILL COMPOSITION

To isolate the effect of compositional structure while holding problem length constant, we analyze performance on $n = 4$ puzzles. Our findings reveal a clear difficulty hierarchy that points to a "lookahead bottleneck" as a fundamental challenge for autoregressive models.

Figure 4 ($n = 4$ column) shows that balanced patterns ($[2] \circ [2]$), which decompose a problem into two equal subtasks, are mastered far more reliably than unbalanced ones. Crucially, right-heavy patterns ($[1] \circ [3]$) are substantially harder to learn than left-heavy ($[3] \circ [1]$) ones. To generate a solution for a right-heavy pattern like $A/(B - C/D)$, the model must commit to the root operator ($\div$) before generating the complex, three-term subroutine it applies to. This need to plan ahead for a complex subsequent operation is the primary failure mode.

Table 2 quantifies this bottleneck across all model sizes: at the same depth of 3, $[1] \circ [3]$ generally underperforms $[3] \circ [1]$, especially with the root operator $\div$ and with smaller models.

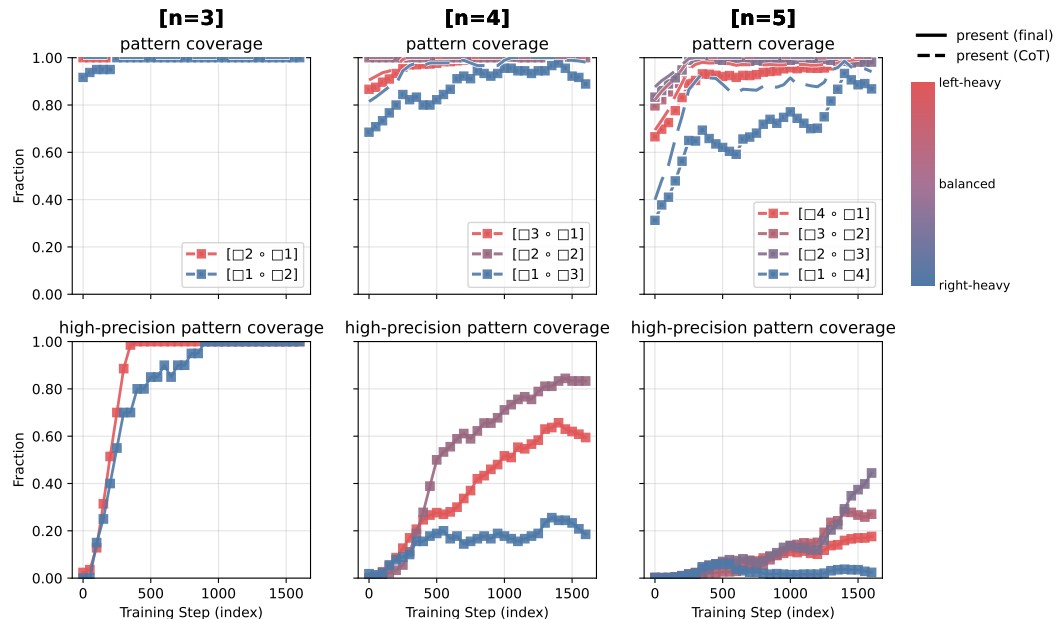

Figure 4: **Compositional structure (not input length) determines difficulty.** Within $n = 4$, balanced patterns ($[2] \circ [2]$; purple) are discovered (top row) and mastered more reliably (bottom row) than the unbalanced ones (red/blue). Balanced patterns have shallow trees, with solution depth of 2 equal to that of $n = 3$ puzzles. Even when controlling for depth, right-heavy patterns ($[1] \circ [3]$; blue) are harder than left-heavy patterns ($[3] \circ [1]$; red). This evidence points to a "lookahead bottleneck," the challenge of committing to a solution ahead of a complex subroutine. The entire structural hierarchy persists for $n = 5$. Results shown for a representative run of `Qwen2.5-1.5B`. See Appendix G for equivalent plots for all other runs.

### 4.4 RL ENABLES SYNTHESIS OF NOVEL PATTERNS, PROVIDING EVIDENCE FOR COMPOSITIONAL GENERALIZATION

We next test for the strongest form of generalization: whether RL can enable a model to synthesize a novel reasoning pattern it has never been explicitly trained on. We designed a rigorous held-out experiment, removing the $n = 3$ pattern $A/B + C$ and its entire family of $n = 4$ derivatives from the training dataset (e.g., $A/B + C + D$, $A/(B + C) + D$). See Appendix C.4 for the full list.

The results provide evidence for compositional generalization. `Qwen2.5-1.5B` trained on this dataset successfully learns to generate the held-out patterns despite zero training exposure (Figures 5, 24 and 25). The learning dynamics also reveal: coverage first emerges on the simpler $n = 3$ subpattern before the model learns to compose it into its more complex $n = 4$ dependent forms. This demonstrates that the model is not merely matching seen templates but is actively reusing learned substructures to assemble novel operator-tree shapes, fulfilling our definition of *compositional generalization*.

### 4.5 ISOLATING THE EFFECT OF RL

We additionally compare with Supervised Fine-tuning (SFT). To prepare the answer column, we replace the placeholder symbols in the desired pattern with the correct list of numbers. For the optional chain of thought text, we 1) randomly select an integer $k \in [2, n]$ based on the puzzle size $n$; 2) for each of $k$ incorrect attempts, randomly choose a pattern different from the desired pattern and replace the placeholder symbols with a randomly shuffled list of the input numbers; 3) evaluate the expression and append "`(Incorrect)`"; 4) append the correct expression and append "`(Correct).`" The CoT is explicitly designed to be random to remove any bias from hand-coded heuristics. See Appendix B.2 for more details on the training and Appendix E for example prompts.

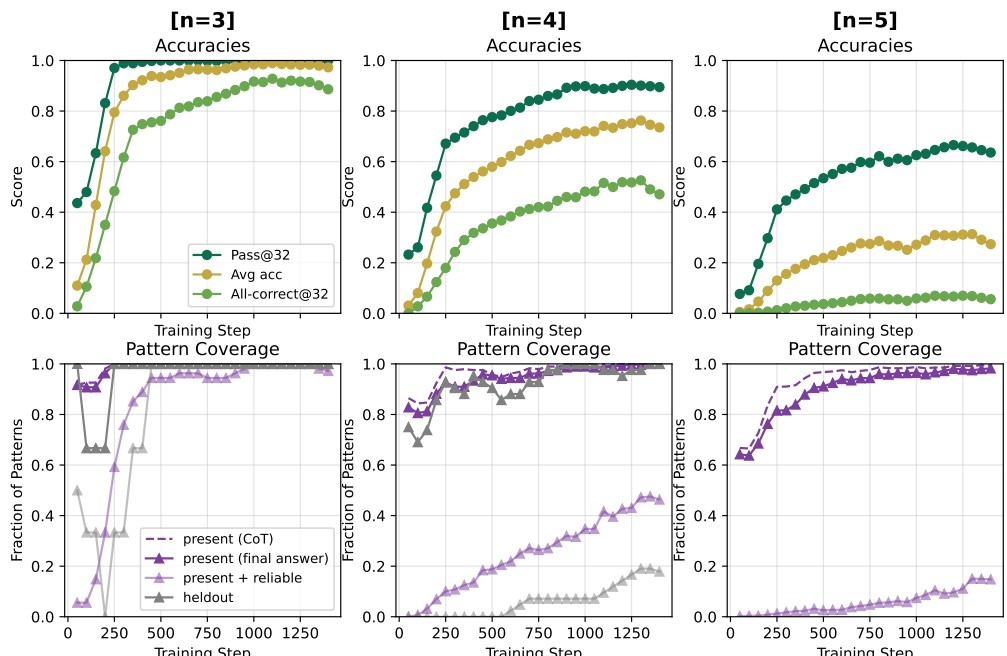

Figure 5: **Generalizes to entire families of held-out compositional patterns.** To evaluate compositional generalization, we removed an entire family of related patterns from the training set: the base $n = 3$ structure $A/B + C$ and all of its $n = 4$ extensions (e.g., $A/B + C + D$, $A/(B + C) + D$). The model successfully recovers these unseen patterns. Coverage first emerges on the held-out $n = 3$ subpattern before generalizing to its more complex $n = 4$ dependents, matching the typical learning hierarchy where presence precedes precision. This result demonstrates that the model can reuse learned substructures to assemble novel operator-tree shapes it has never been explicitly trained on.

**Final accuracies.** Table 3 reports the final accuracies of `Qwen2.5-1.5B` trained with SFT. The model performs better without the CoT, most likely because including the random patterns confuses the model. But, SFT without CoT still underperforms RL and struggles to generalize to $n = 5$, with Average Accuracy less than 10%. This suggests that the exploratory nature of RL is more effective than the imitation-based approach of SFT, which struggles to generalize especially when the provided data is too simple or poorly structured. Nonetheless, the learning dynamics of SFT can isolate the order of skill acquisition without access to the model's own exploration and skill composition.

**SFT exhibits different learning dynamics by pattern structure.** The model trained without CoT fails to reliably learn most $n = 4$ patterns. To probe the order in which the model learns the patterns to an intermediate level, we instead relax the threshold of high-precision to 50% (Figure 9). Even though the coverage of patterns for $n = 5$ patterns decrease in a similar order as in RL: right-heavy > left-heavy > balanced, the model instead masters $n = 4$ patterns (in-distribution) in the exact opposite order: right-heavy < left-heavy < balanced. This highlights that the clear hierarchy of difficulty caused by the subtree structure is inherent to the RL training.

## 5 ABLATIONS

In Appendix C, we run ablations on the different part of the training pipeline. Small models often find a formatting-only shortcut, but increasing the group size initially mitigates the instability (Appendix C.1). Normalizing by the standard deviation in the group advantage leads to longer responses and a lower final performance (Appendix C.2); increasing the number of rollouts may improve stability but leads to a lower final performance (Appendix C.3); models require harder and a diverse set of patterns to escape the formatting-only shortcut (Appendix C.4); models trained without a random subset of patterns can still generalize to unseen patterns and larger pattern sizes (Appendix C.4); training with PPO may improve All-correct@32 but hurts Pass@32 (Appendix C.5).

Table 3: **Final performance of SFT-trained models.** All-correct@32 (All), Average Accuracy (Avg), and Pass@32 (Pass) on $n=3, 4$ (trained) and $n=5$ (held out). We present the performance of the base model and the GRPO-trained model as a basis for comparison.

| | $n=3$ | | | $n=4$ | | | $n=5$ | | |
|---|---|---|---|---|---|---|---|---|---|
| Training | All | Avg | Pass | All | Avg | Pass | All | Avg | Pass |
| Base Model | 0.00 | 0.02 | 0.42 | 0.00 | 0.00 | 0.00 | 0.00 | 0.00 | 0.02 |
| GRPO | 0.98 | 1.00 | 1.00 | 0.63 | 0.83 | 0.94 | 0.13 | 0.36 | 0.68 |
| SFT+No CoT | 0.31 | 0.65 | 0.92 | 0.05 | 0.38 | 0.82 | 0.00 | 0.09 | 0.33 |
| SFT+Random CoT | 0.01 | 0.15 | 0.59 | 0.00 | 0.03 | 0.25 | 0.00 | 0.00 | 0.04 |

## 6 RELATED WORKS

**Skill composition in LLMs** Arora and Goyal (2023) establish a theoretical foundation for the emergence of skill composition in LLMs, and Yu et al. (2024) benchmark the ability of the models to compose skills. Regarding how to elicit skill composition, previous works focus mainly on pre-training and supervised fine-tuning: Chen et al. (2023) propose optimizing data ordering, whereas other works investigate in-context prompting (Chen et al., 2024) and supervised fine-tuning (Zhao et al., 2024). This work extends the analysis of skill composition to the RL post-training stage.

**Length and compositional generalization** Previous works (Anil et al., 2022; Zhou et al., 2024; Lee et al., 2025) study the ability of LLMs to generalize to a sequence length longer than seen during training. More broadly, recent papers DeepSeek-AI et al. (2025); Setlur et al. (2025) claim that RL helps models generalize to more difficult problems by encouraging more attempts, considering length as a proxy for difficulty. Sun et al. (2025) study the ability of RL to teach LLMs to compose skills, but they consider the number of skills combined as a proxy for task complexity, still conflating length and compositional generalization. Concurrent with our work, Yuan et al. (2025) also show that RL can induce compositional ability and report the need for a metric beyond pass@k, but their definition is limited to compositional depth, which again coincides with length. Our work instead proposes a framework to analyze the structure behind each skill as an abstract tree, which allows us to diagnose how the complexity of a task is determined by the compositional structure, not just its length or depth.

**Sharpening vs discovery and pass@k.** Previous works (Yue et al., 2025; Dang et al., 2025) often use the metric of pass@k to label the effect of post-training as either *sharpening* (Huang et al., 2025)—concentrating probability on pre-existing good paths—or *discovery*—acquiring new knowledge. While useful, trajectory-level metrics like pass@k are too coarse to provide detailed insight into model behavior. Our work focuses on the learning dynamics, which are compatible with both possibilities. Instead of concluding whether RL is introducing a new type of reasoning or resurfacing it, we focus more on skill composition: *how* and *in which order* it happens.

**COUNTDOWN and the game of 24.** Prior works (Yao et al., 2023; Gandhi et al., 2024; Herr et al., 2025; Ni et al., 2025) consider COUNTDOWN, and its variant Game of 24, as a testbed for model reasoning due to its clean structure. They mainly focus on optimizing inference-time search or training on heuristics-based search traces, but these approaches can make it difficult to identify which skills acquired versus which are a part of the search algorithm / heuristics. By contrast, our work focuses on *natural RL* on COUNTDOWN—no handcrafted search heuristics or specialized decoding—so the only signal is the natural task reward. This design lets us attribute behavioral changes directly to RL and to analyze them with the pattern framework.

**Design choices in RL training.** RL training is sensitive to algorithmic choices. The loss function in GRPO (Shao et al., 2024) involves multiple terms, including group-level advantage normalization, KL regularization, and gradient clipping. Follow-up works (Liu et al., 2025; Yu et al., 2025; Shrivastava et al., 2025) analyze the effect of these choices; for example, arguing that removing standard-deviation

normalization can improve stability and avoid biases toward easy prompts. Our work complements these works by providing additional empirical observations on the effect of the design choices in RL.

**Skill composition in classical RL**    In classical RL, a skill is defined in the framework of options (Sutton et al., 1999; Barto and Mahadevan, 2003). These works view skill composition as a sequential application of individual skills, which aligns with our definition of length generalization. On the other hand, other works (Todorov, 2009) define skill composition as the ability to learn a single behavior that can solve multiple objectives via a combined value function, which would be considered compositional generalization in our definition. While these frameworks are defined at the MDP level, the present work follows the norm in the LLM literature and defines skills at the task level, defined on the generated outputs of the model. See Mendez and Eaton (2023) for a survey of skill compositions in classical RL.

## 7    DISCUSSION AND FUTURE WORK

Our work studies the acquisition of skill composition during RL. On natural data, compositional complexity is confounded with length, making it impossible to isolate true failure modes. A synthetic testbed is therefore methodologically essential to control for these confounders and identify drivers of reasoning failure. Thus, the COUNTDOWN task introduces a formal lens through which to analyze the acquisition of compositional skills. By representing solutions as computational trees, we can disentangle the effects of a problem's length from its underlying compositional structure. This decomposition reveals a fundamental insight: the primary barrier to learning complex reasoning are not length-based but structural. Models do not fail simply because a task is long; they fail when its compositional form creates specific bottlenecks.

Our analysis reveals that problems decomposable into balanced sub-problems are learned most readily. In contrast, those requiring deep-sequential commitments (particularly "right-heavy" structures that demand significant lookahead before executing a complex subroutine) constitute a fundamental bottleneck for autoregressive models; crucially, these right-heavy structures are often harder than left-heavy ones, despite having the same compositional depth. This demonstrates that a task's compositional structure, rather than merely its size, is what dictates learnability. This finding should be understood not as an artifact of our task, but as highlighting a plausible explanation of where difficulty in extending beyond the reasoning boundary arises. Our results suggest that the inherent challenge is tied to the structure of skill composition itself, not to incidental features of the data.

This structural view of difficulty likely extends to other domains–such as program synthesis, multi-step tool use, and theorem proving–where solutions can be represented as computation or proof trees. In such settings, structural properties (balance, depth, and lookahead/sequential dependencies) may predict difficulty. We argue that the overall sample complexity of learning skill composition is not monolithic; instead, it is dominated by the challenge of mastering these structurally-hard instances.

This insight points to curriculum design as a promising direction for future work. We hypothesize that, once RL post-training yields signs of length and some compositional generalization, dynamically augmenting the training distribution with a minimal set of structurally hard instances may further advance the model's capability frontier. Verifying this hypothesis and developing such curricula remain a crucial direction for future work.

## 8    LIMITATIONS

The scope of the paper is limited to COUNTDOWN; we do not evaluate on other reasoning or compositional tasks. We consider generalization only to a larger puzzle size $n$ or to a selected held-out set of patterns; a more careful design of held-out strategy will be useful to test the limit of the generalization capability. Finally, the paper trains on a random mix of a balanced dataset; we do not consider any data curriculum, which may further accelerate the rate of learning or reveal other learning dynamics of RL.

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

## A    EXPERIMENTAL SETUP (MORE DETAILS)

### A.1    EXPRESSIONS TO PATTERNS

Here are the heuristics (in plain English) to map an expression to its canonical form.

- Remove any unnecessary parentheses. For example, map $(A + B) + C$ to $A + B + C$.
- If possible, use $+$ and $\times$ instead of $-$ and $/$. For example, map $A - (B - C)$ to $A - B + C$.
- For commutative operations ($+$ and $\times$), the operands are sorted in decreasing order of the "weight" of the operand (i.e., the number of symbols involved in the intermediate term). For example, $(A + B) \times C$ is preferred over $A \times (B + C)$.
- If in the same intermediate level of computation, $+$ and $\times$ should appear before $-$ and $/$. For example, $A + B - C$ is preferred over $A - B + C$. For example, $(A + B) \times (A - B)$ is preferred over $(A - B) \times (A + B)$.

The full list of mapping is given in Tables 13 to 16. In Appendix F, we show that the choice of the canonicalization scheme does not affect the main findings of the paper.

### A.1.1    A TREE VIEW OF EXPRESSIONS AND PATTERNS

Each **expression** can be viewed as a binary tree, with each node representing an operation and each leaf node representing a symbol. To define congruence between equivalent expressions, we convert them to a signed $n$-ary tree:

- Each node now represents a chain of addition/subtraction operations ($\oplus$) or a chain of multiplication/division operations ($\otimes$). Each node can now connect to any number of operands that are being computed in that chain. For example, $A + B + C$ is represented with one node with 3 children.
- Each edge is assigned a positive sign if it connects to an operand that is being added/multiplied in the chain or a negative sign if connects to an operand that is being subtracted/divided in the chain.
- To convert from a binary tree to a $n$-ary tree:
    - For nodes representing addition/multiplication, mark both edges as positive. For nodes representing subtraction/division, mark the left edge as positive but the right as negative.
    - Recursively merge any pair of parent/child if they belong in the same chain. The sign of the edges of the child node will be flipped if the edge between the parent and the child was negative.

If the binary trees have the same representation in the signed $n$-ary tree, they are considered equivalent. The **pattern** (canonical form) can be retrieved as follows:

- At any level, sort the children by
    1. sign of the edge between the shared parent and the child (positive comes first);
    2. the weight of the subtree rooted at the child (heavy comes first);
    3. the number of positive edges to the grandchildren (more positive edges comes first).
- Do an in-order traversal of the tree. Print the first node as $A$, the second node as $B$, and so forth. Include parentheses only if a child subtree has weight $> 1$ and is rooted in a $\oplus$ node.

### A.1.2    NUMBER OF EXPRESSIONS AND PATTERNS

Given $n$, the number of distinct **expressions** is $C_{n-1} \times 4^{n-1}$ where $C_n$ is the Catalan number. There are $C_{n-1}$ ways to assign the order of $n - 1$ operations (equivalently, choose the structure of the binary tree). For example, when $n = 4$, there are 5 ways: $((AB)C)D$, $(A(BC))D$, $(AB)(CD)$, $A((BC)D)$, and $A(B(CD))$. There are additionally $4^{n-1}$ different ways to choose the operations.

The number of distinct **patterns** $T_n$ is equal to the number of trees with $n$ leaf nodes with the following constraints: 1) any child of a $\oplus$ node must be a leaf node or a $\otimes$ node (and vice versa); 2)

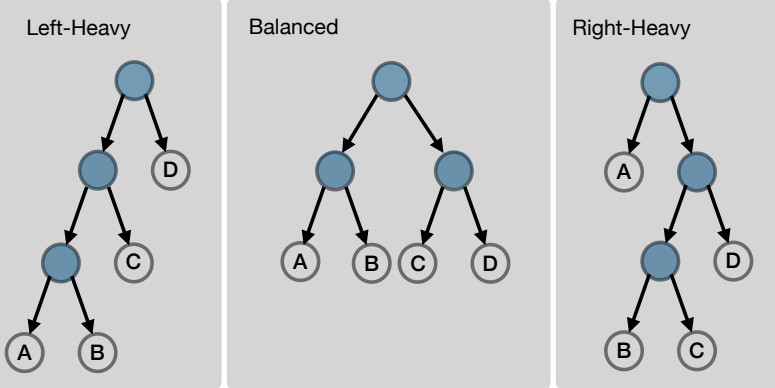

Figure 6: **Grouping canonical patterns by tree structure** ($n = 4$). Internal (blue) nodes are operators and leaves $(A, B, C, D)$ are operands. We categorize patterns by the relative size of the root's subtrees: left-heavy, balanced, or right-heavy.

no node shall have all negative edges. We let $A_n, M_n$ respectively denote the number of such trees with a $\oplus$ root node and a $\otimes$ root node. We let $A_1 = 1$ (for the single leaf $A$) and $M_1 = 0$. For $n \geq 2$, $T_n, A_n, M_n$ can be computed with the following recursive definition.

$$
\begin{cases}
A_n = n + \displaystyle\sum_{\substack{p \in \bigcup_{2 \leq k < n} P(n,k)}} \left[ (g_1 + 1) \left( \prod_{j=1}^{m_p} \binom{g_{p,j} + 2M_{d_{p,j}} - 1}{g_{p,j}} \right) - \left( \prod_{j=1}^{m_p} \binom{g_{p,j} + M_{d_{p,j}} - 1}{g_{p,j}} \right) \right] \\[2em]
M_n = n + \displaystyle\sum_{\substack{p \in \bigcup_{2 \leq k < n} P(n,k)}} \left[ (g_1 + 1) \left( \prod_{j=1}^{m_p} \binom{g_{p,j} + 2A_{d_{p,j}} - 1}{g_{p,j}} \right) - \left( \prod_{j=1}^{m_p} \binom{g_{p,j} + A_{d_{p,j}} - 1}{g_{p,j}} \right) \right] \\[2em]
T_n = A_n + M_n
\end{cases}
$$

where

- $P(n, k)$ is the unordered set of partitions of $n$ into a sum of $k$ positive integers
- Each partition $p \in \displaystyle\bigcup_{2 \leq k < n} P(n, k)$ is written as

$$
p = \overbrace{(d_{p,1} + d_{p,1} + \cdots + d_{p,1})}^{g_{p,1} \text{ times}} + \cdots + \overbrace{(d_{p,m_p} + \cdots + d_{p,m_p})}^{g_{p,m_p} \text{ times}} + \overbrace{(1 + \cdots + 1)}^{g_1 \text{ times}}
$$

where

  - $m_p$ is the number of distinct part sizes in $p$ (other than 1)
  - $d_{p,j} > 1$ are the distinct part sizes in $p$
  - $g_{p,j}$ are their corresponding multiplicities, and $g_1$ is the multiplicity for 1

The first $n$ corresponds to assigning $\{0, 1, \cdots, n - 1\}$ negative edges to $n$ leaf nodes directly connected to the root node (which we assume to be $\oplus$ without loss of generality). Then for any partition $p \in P(n, k)$ where $2 \leq k < n$, any $d_{p,j} > 1$ corresponds to a $\otimes$ child node with weight $d_{p,j}$, whereas $d_{p,j} = 1$ corresponds to a leaf node directly connected. If $d_{p,i} \neq d_{p,j}$, then the corresponding $\otimes$ child nodes are distinguishable, so we can independently count the number of choices for each child node.

For each $d_{p,j} > 1$, there could be multiple $\otimes$ child nodes that have the same weight and can have one of $M_{d_{p_j}}$ shapes. If they have the same shape, they are indistinguishable. For each $i = 1, 2, \cdots, M_{d_{p,j}}$, let $x_i \geq 0$ denote the number of the child nodes that have the corresponding shape such that

$$
x_1 + x_2 + \cdots + x_{M_{d_{p,j}}} = g_{p,j}
$$

For each $x_i$, there are $x_i + 1$ distinguishable ways to assign a sign to each child node ($\{0, 1, \cdots, x_i\}$ negative edges). Therefore, we apply the stars and bars formula to get:

$$\sum_{\substack{x_1 + \cdots + x_{M_{d_{p,j}}} = g_{p,j} \\ x_i \geq 0 \ \forall i}} (x_1 + 1)(x_2 + 1) \cdots (x_{M_{d_{p,j}}} + 1)$$

$$= \sum_{\substack{y_1 + \cdots + y_{M_{d_{p,j}}} = g_{p,j} + M_{d_{p,j}} \\ y_i \geq 1 \ \forall i}} y_1 y_2 \cdots y_{M_{d_{p,j}}}$$

$$= \binom{(g_{p,j} + M_{d_{p,j}}) + M_{d_{p,j}} - 1}{2 M_{d_{p,j}} - 1}$$

$$= \binom{g_{p,j} + 2 M_{d_{p,j}} - 1}{g_{p,j}}$$

When $d = 1$, the leaf children are all indistinguishable and there are $g_1 + 1$ ways to assign a sign to each leaf. We finally need to subtract the case where we assigned a negative edge to all possible $g_{p,j}$ child nodes of all possible weight of $d_{p,j}$. For each choice of $x_i$, there is exactly 1 way to assign a negative sign to all child nodes. Again, by the stars and bars formula, we get

$$\sum_{\substack{x_1 + \cdots + x_{M_{d_{p,j}}} = g_{p,j} \\ x_i \geq 0 \ \forall i}} 1 = \binom{g_{p,j} + M_{d_{p,j}} - 1}{g_{p,j}}$$

## A.2 DATASET

### A.2.1 WHY IS THE EXISTING DATASET LACKING?

Pan et al. (2025b) do not release the exact generation code for their dataset. So instead, we sample 10000 examples from the training split of the dataset and count the number of examples that can be solved with a given pattern (if one example can be solved with multiple patterns, increment the count of all such patterns). 18 of the 114 patterns never appear in the dataset (i.e., none of the examples can be solved with the patterns), and some patterns occur much more often than others (Figure 7). In the test split (the last 1000 examples), there are 36 out of 114 patterns that never appear.

One reason behind the imbalance is that almost half of the dataset consists of examples for $n = 3$, even though there are only 18 patterns for $n = 3$ compared to 96 for $n = 4$. Another explanation is that the requirement that the target number is an integer within $[1, 99]$ biases towards addition and subtraction—multiplication would frequently cause the expression to be larger than the bound; division would frequently make the final value non-integer.

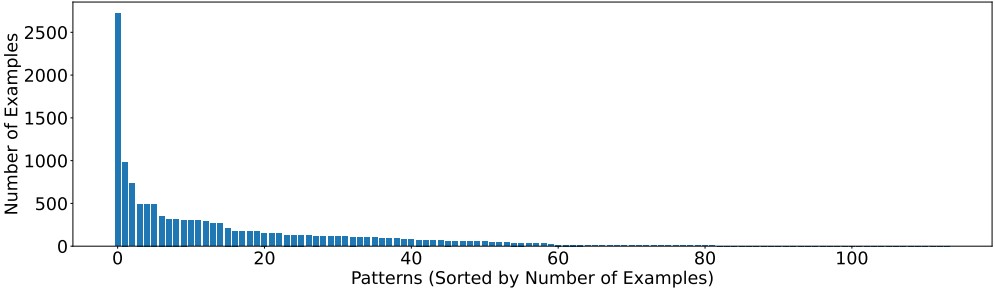

Figure 7: **Number of examples per pattern in Pan et al. (2025b).** 18 out of 114 patterns do not appear in a randomly selected 10000 examples from the dataset. Few patterns (mostly from $n = 3$) appear disproportionally frequently.

Two other codebases that provide the generation code for Countdown (Gandhi et al., 2024; Stojanovski et al., 2025) wrongly assume that intermediate expressions cannot evaluate to non-integer values. This incorrectly removes puzzles like $8 \div (3 - 8 \div 3) = 24$ which are very difficult even for humans (4nums.com, 2025) and should be used to test against models.

### A.2.2 WHAT HAPPENS WHEN TRAINING ON THE EXISTING DATASET?

We train `Qwen2.5-3B` on the dataset by Pan et al. (2025a) with the same set of hyperparameters and evaluate on both our and their held-out data. Since we do not explicitly check if our held-out data is in their training data, the results on our held-out data may be slightly higher than expected. In Figure 8, we recreate Figure 3.

Even though the model performance seems to smoothly improve, the final performance is much lower than training on our balanced dataset, which stems from the model failing to reliably learn most $n = 4$ patterns. This suggests that the diversity of the data and the presence of challenging patterns helps the model identify reusable components. Additionally, many patterns are not present when evaluating on the held-out data from Pan et al. (2025a), precisely because none of the puzzles test those patterns. This points to another deficiency in the existing datasets for COUNTDOWN that test simple additive skills, rather than a well-rounded portfolio of arithmetic reasoning.

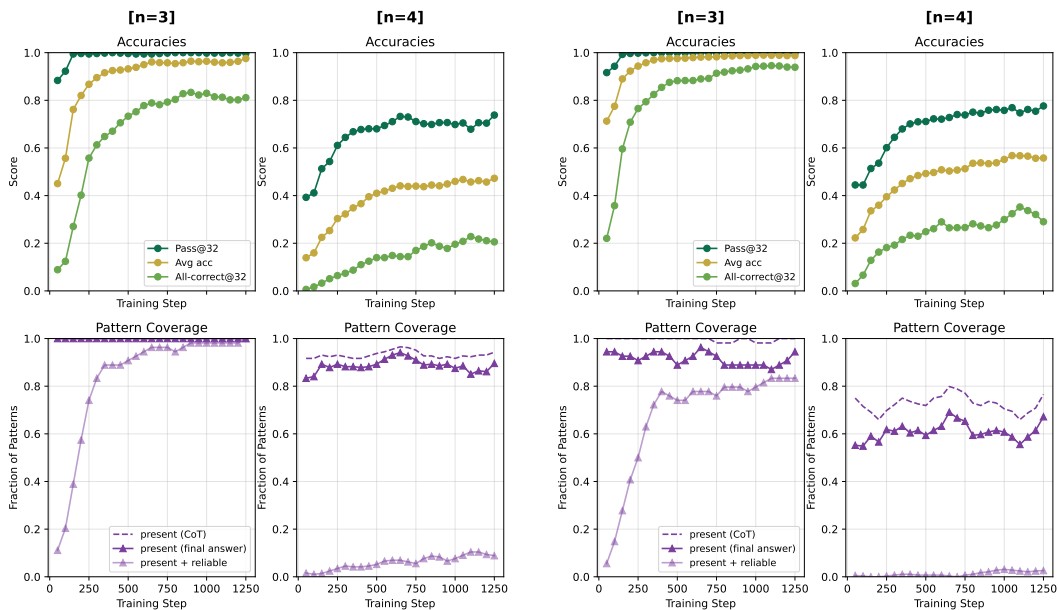

Figure 8: **Learning dynamics at a glance (Qwen2.5-3B)** trained on dataset from Pan et al. (2025a). **(Left)** evaluated on our dataset; **(Right)** evaluated on the held-out data from Pan et al. (2025a). Even though model accuracy smoothly increases (top), models do not reliably learn most $n = 4$ patterns and fails to generate at least 10% of $n = 4$ patterns (bottom). The held-out data from Pan et al. (2025a) do not include 36 possible patterns and fail to test the models on the corresponding skills.

### A.2.3 CREATING A BALANCED DATASET

To create our balanced dataset, we first choose a pattern and then plug in $n$ numbers that are sampled independently and uniformly at random from $[1, 99]$. If the output of the expression is an integer in $[1, 99]$, we include it in our dataset. *For each pattern*, we repeat this process until we have 4000 distinct examples or have sampled 10 million combinations. There are only 15 patterns (4 from $n = 3$ and 11 from $n = 4$) where we sample less than 4000 examples. The exact numbers are given in Table 4. For each pattern, we select the first 10 examples as heldout examples. In total, we have 418619 examples for training and 1140 examples for testing.

For evaluation on $n = 5$, we similarly sample 10 examples for each of 558 patterns and for $n = 6$, 1 example for each of 4328 patterns.

### A.3 TRAINING

We train with batch size and mini batch size 256, prompt and response length 1024, learning rate $10^{-6}$, and KL coefficient $10^{-3}$, where the KL divergence is computed with the $k_3$ estimator (low

Table 4: **Number of training examples per pattern.** Any pattern not listed has 4000 examples.

| Pattern | Number of Examples | Pattern | Number of Examples |
|---|---|---|---|
| $(A + B) \times C$ | 3333 | $A \times B \times C$ | 315 |
| $A/(B + C)$ | 3905 | $A/B/C$ | 791 |
| $(A \times B + C) \times D$ | 2314 | $(A + B) \times C \times D$ | 927 |
| $A \times B \times C \times D$ | 188 | $A/(B \times C + D)$ | 2704 |
| $A/(B + C)/D$ | 1058 | $A/B/C/D$ | 264 |
| $A/B - C \times D$ | 2546 | $(A + B) \times (C + D)$ | 647 |
| $A/(B + C) - D$ | 932 | $A/B/C - D$ | 1033 |
| $(A/B - C)/D$ | 2802 | - | - |

variance) (Schulman, 2020). Following Liu et al. (2025), we do not normalize by the standard deviation in the GRPO advantage computation. We use $n = 4$ rollouts except during a warmup phase for 1.5B and 3B models (Appendix C.1). For the PPO experiments in Appendix C.5, we use a critic learning rate of $10^{-5}$ and PPO mini bach size of 64. For all other hyperparameters, we use the default values from the `verl` package version 0.5.0.dev0 (Sheng et al., 2024). See Appendix E for the prompt template used.

# B MAIN RESULTS (CONTINUED)

## B.1 ADDITIONAL TABLES

1.5B/3B models smoothly improve performance throughout training (1635 steps), whereas the 7B model reaches peak performance before 1 epoch. In the main paper, we reported the performance of the checkpoints right before the training reward declines (1350/950/650 steps respectively for each seed). For completeness, we also provide the performance of the final checkpoints of the 7B model in Tables 5 and 6. The 7B model can eventually improve on right-heavy patterns ($[1] - [3]$ or $[1] \div [3]$) but at the cost of sacrificing performance on balanced and left-heavy patterns. Additionally, we report the standard deviation when averaging across random seeds.

In Table 5, we also report the performance of the base models before GRPO training. Note that the Pass@32 for these models is very close to $1 - (1 - p)^{32}$ where $p =$ Average Accuracy, suggesting that the models are randomly guessing the output. At the same time, within each output, the model tends to randomly guess multiple expressions, until it runs out of the generation budget. Since there is only a small subset of expressions that are valid for $n = 3, 4$ (up to $18 \times 3! = 108$ for $n = 3$ and $96 \times 4! = 2304$), the random baseline for the base models is quite high (assuming 10 guesses per output, $p \approx 0.1$ for $n = 3$ and $p \approx 0.05$ for $n = 4$). The observed performance of the base models is not far from the random baseline.

Table 5: **Performance across model size (Full version).** Averaged across all successful seeds (**final** (F) checkpoints for 1.5B/3B and **best** (B) and **final** (F) checkpoints for 7B). All-correct@32 (All), Average Accuracy (Avg), and Pass@32 (Pass) on $n=3, 4$ (trained) and $n=5, 6$ (held out). Models learn $n = 3, 4$ patterns almost perfectly and generalize to larger puzzle sizes $n = 5, 6$, but the gap between Pass@32 and Average Accuracy increases with $n$, signaling that the generalization is not perfect.

| Model | All | Avg | Pass | All | Avg | Pass |
|---|---|---|---|---|---|---|
| | | $n=3$ | | | $n=4$ | |
| 1.5B | 0.00 | 0.02 | 0.42 | 0.00 | 0.00 | 0.00 |
| 1.5B (F) | $0.97 \pm 0.01$ | $1.00 \pm 0.00$ | $1.00 \pm 0.00$ | $0.60 \pm 0.09$ | $0.82 \pm 0.02$ | $0.93 \pm 0.00$ |
| 3B | 0.00 | 0.05 | 0.66 | 0.00 | 0.01 | 0.20 |
| 3B (F) | $0.93 \pm 0.01$ | $0.99 \pm 0.01$ | $1.00 \pm 0.00$ | $0.56 \pm 0.06$ | $0.81 \pm 0.03$ | $0.94 \pm 0.01$ |
| 7B | 0.00 | 0.10 | 0.88 | 0.00 | 0.02 | 0.36 |
| 7B (B) | $0.99 \pm 0.01$ | $1.00 \pm 0.00$ | $1.00 \pm 0.00$ | $0.61 \pm 0.07$ | $0.86 \pm 0.03$ | $0.98 \pm 0.01$ |
| 7B (F) | $0.95 \pm 0.02$ | $0.99 \pm 0.00$ | $1.00 \pm 0.00$ | $0.56 \pm 0.03$ | $0.80 \pm 0.06$ | $0.92 \pm 0.06$ |
| | | $n=5$ | | | $n=6$ | |
| 1.5B | 0.00 | 0.00 | 0.02 | 0.00 | 0.00 | 0.01 |
| 1.5B (F) | $0.13 \pm 0.05$ | $0.36 \pm 0.03$ | $0.68 \pm 0.00$ | $0.00 \pm 0.00$ | $0.09 \pm 0.00$ | $0.36 \pm 0.00$ |
| 3B | 0.00 | 0.00 | 0.03 | 0.00 | 0.00 | 0.00 |
| 3B (F) | $0.12 \pm 0.02$ | $0.39 \pm 0.01$ | $0.68 \pm 0.02$ | $0.02 \pm 0.00$ | $0.15 \pm 0.00$ | $0.40 \pm 0.00$ |
| 7B | 0.00 | 0.00 | 0.08 | 0.00 | 0.00 | 0.02 |
| 7B (B) | $0.15 \pm 0.02$ | $0.46 \pm 0.04$ | $0.75 \pm 0.03$ | $0.02 \pm 0.00$ | $0.15 \pm 0.00$ | $0.39 \pm 0.00$ |
| 7B (F) | $0.10 \pm 0.02$ | $0.35 \pm 0.05$ | $0.61 \pm 0.10$ | $0.01 \pm 0.00$ | $0.12 \pm 0.00$ | $0.36 \pm 0.00$ |

Table 6: **Per-pattern precision averaged by pattern structure on** $n{=}4$ **(Full version).** Averaged across all successful seeds (**final** checkpoints for 1.5B/3B and **best** and **final** checkpoints for 7B). Averaged by (a) root operator; (b) pattern with root operator $-$; (c) pattern with root operator $\div$. Models struggle more on $-, \div$ as the root operator. Shapes in the form of $[1] - [3]$ or $[1] \div [3]$ are particularly challenging. Within each column, the highest number is **boldfaced** and the second highest number is underlined.

(a) by root operator

| Operator | 1.5B | 3B | 7B (best) | 7B (final) |
|---|---|---|---|---|
| $+$ | $0.86 \pm 0.08$ | $0.85 \pm 0.01$ | **$0.91 \pm 0.01$** | $0.83 \pm 0.03$ |
| $-$ | $0.83 \pm 0.09$ | $0.80 \pm 0.04$ | $0.86 \pm 0.06$ | $0.76 \pm 0.11$ |
| $\times$ | **$0.90 \pm 0.01$** | **$0.86 \pm 0.03$** | **$0.91 \pm 0.01$** | **$0.84 \pm 0.08$** |
| $\div$ | $0.75 \pm 0.04$ | $0.78 \pm 0.06$ | $0.83 \pm 0.04$ | $0.81 \pm 0.08$ |

(b) by subtree shape ($-$)

| Shape | 1.5B | 3B | 7B (best) | 7B (final) |
|---|---|---|---|---|
| $\boxed{3} - \boxed{1}$ | $0.81 \pm 0.13$ | $0.78 \pm 0.03$ | $0.84 \pm 0.04$ | $0.72 \pm 0.12$ |
| $\boxed{2} - \boxed{2}$ | **$0.92 \pm 0.01$** | **$0.89 \pm 0.00$** | **$0.90 \pm 0.07$** | **$0.86 \pm 0.10$** |
| $\boxed{1} - \boxed{3}$ | $0.73 \pm 0.00$ | $0.76 \pm 0.09$ | $0.86 \pm 0.12$ | $0.82 \pm 0.10$ |

(c) by subtree shape ($\div$)

| Shape | 1.5B | 3B | 7B (best) | 7B (final) |
|---|---|---|---|---|
| $\boxed{3} \div \boxed{1}$ | $0.77 \pm 0.05$ | $0.77 \pm 0.05$ | $0.83 \pm 0.05$ | $0.83 \pm 0.05$ |
| $\boxed{2} \div \boxed{2}$ | **$0.88 \pm 0.10$** | **$0.85 \pm 0.00$** | **$0.91 \pm 0.07$** | **$0.89 \pm 0.14$** |
| $\boxed{1} \div \boxed{3}$ | $0.61 \pm 0.03$ | $0.76 \pm 0.12$ | $0.80 \pm 0.17$ | $0.80 \pm 0.05$ |

## B.2 EFFECT OF SUPERVISED FINE-TUNING (SFT) (CONTINUED)

Here, we continue the discussion in Section 4.5.

### B.2.1 TRAINING

We train with batch size 256, prompt and response length 1024, max learning rate $10^{-6}$ with a linear warmup of 0.03 and cosine decay to zero. For all other hyperparmeters, we use the default values from the `verl` package. We train the `Qwen2.5-1.5B` model with and without the random chain of thought text, each with 1 random seed for dataset shuffling.

### B.2.2 LEARNING DYNAMICS

In Figures 9 to 11, we present the learning dynamics plots for the `Qwen2.5-1.5B` model trained with SFT (equivalent to Figure 4).

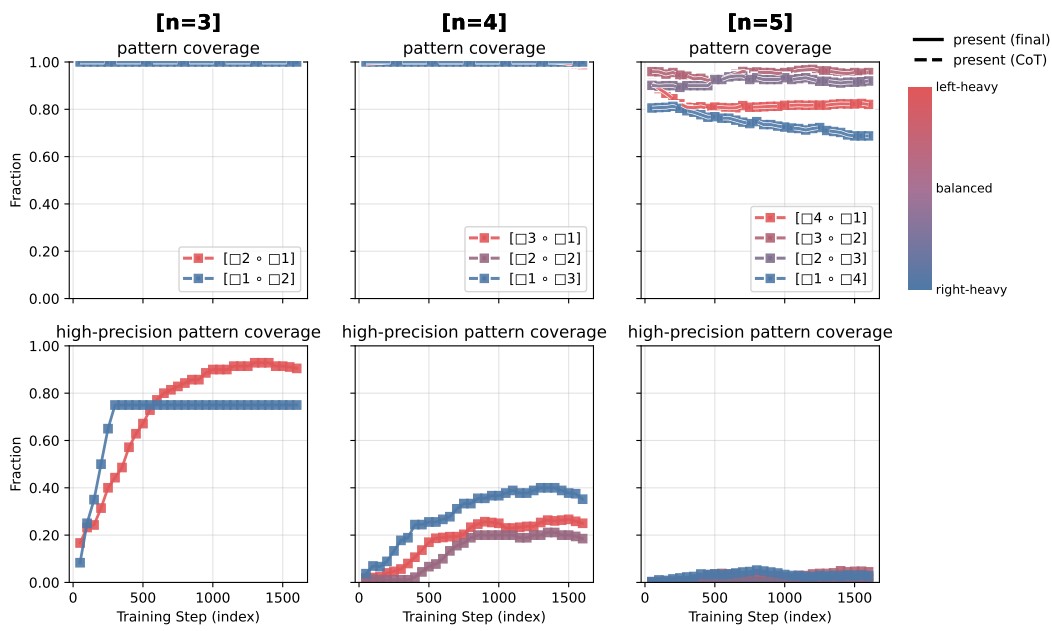

Figure 9: **Learning dynamics at a glance (Qwen2.5-1.5B with SFT)**. Here, the high-precision threshold has been lowered to 50%. Even though the coverage of $n = 5$ patterns (OOD) decrease in the following order: balanced > left-heavy > right-heavy, the model masters $n = 4$ patterns (in-distribution) in the opposite order: right-heavy > left-heavy > balanced. This shows a fundamental difference in the learning dynamics from RL.

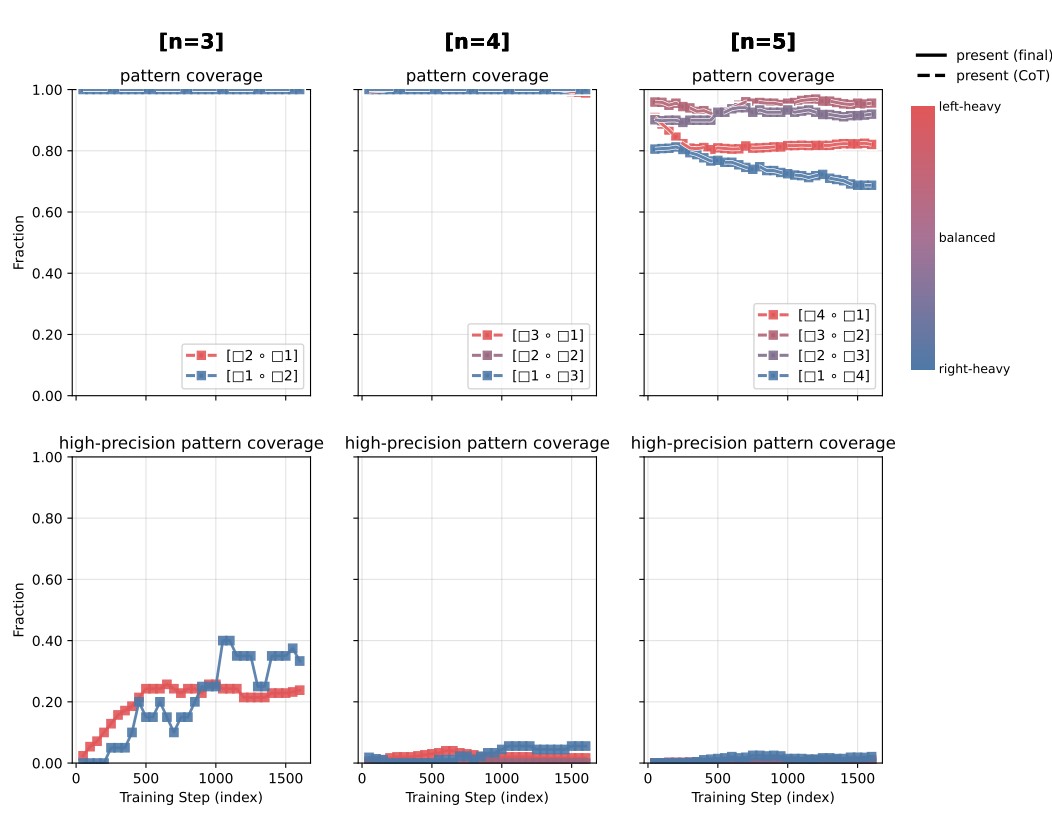

Figure 10: **Learning dynamics at a glance (Qwen2.5-1.5B with SFT)**. The model does not reliably learn any $n = 4$ pattern.

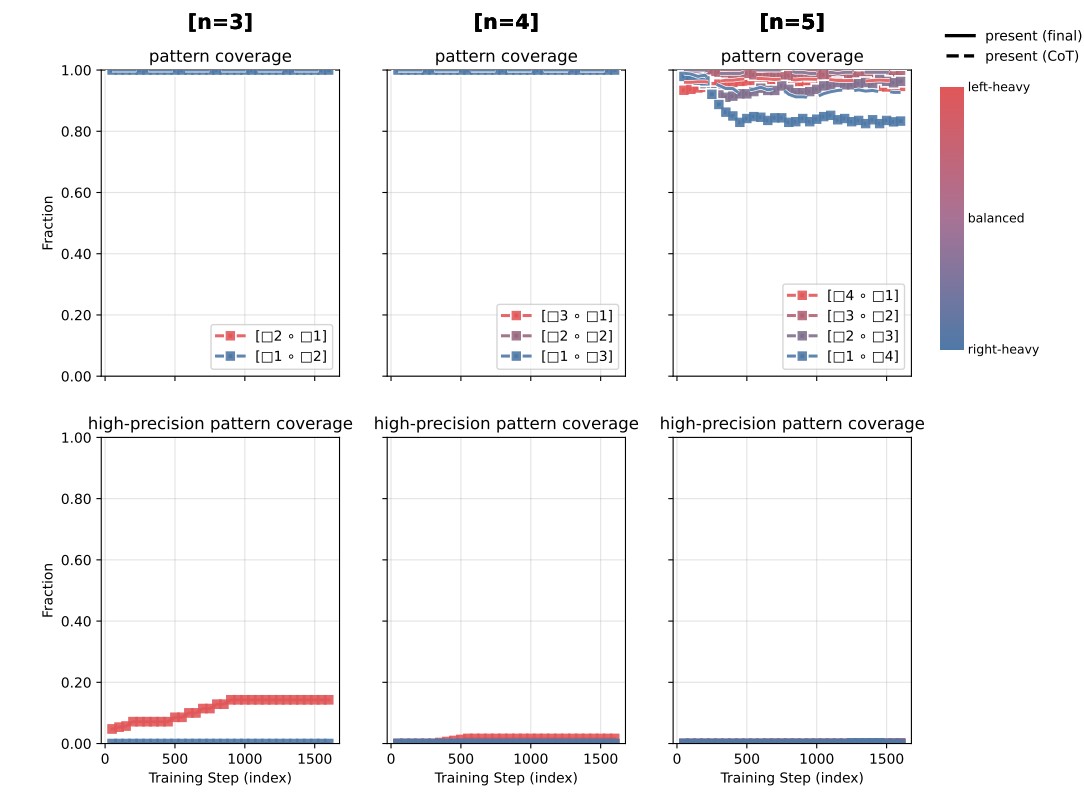

Figure 11: **Learning dynamics at a glance (Qwen2.5-1.5B with SFT+CoT)**. The model does not reliably learn any $n = 4$ pattern.

# C    ADDITIONAL ABLATIONS

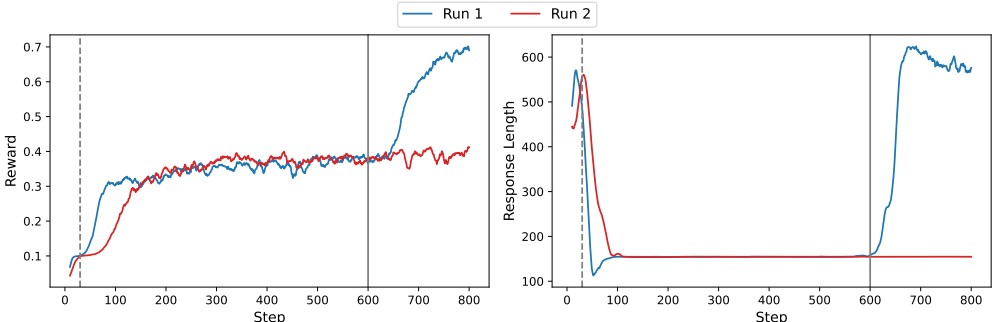

Figure 12: **Qwen2.5-1.5B often learns a formatting-only shortcut: (Left)** training reward; **(Right)** training response length of two distinct training runs. Once the model learns to collect the formatting reward (grey dotted vertical), the model attempts the puzzle without chain of thought, characterized by a sharp decrease in the response length. During some but not all runs (blue), we observe a phase transition (grey solid vertical), where the model starts generating chain of thought again.

## C.1    A FORMATTING-ONLY SHORTCUT, AND HOW TO AVOID IT

Each answer can receive 0.1 points for correct formatting, plus up to 0.9 points for correctness. Smaller models (1.5B and 3B) often exploit the format-only reward first (Figure 12), whereas the larger 7B model does not. This shows that pretraining endows the larger model with more advanced reasoning circuits. For example, if it can collect reward 0.9 with probability $> 0.1$ from the start, then it has no incentive to overfitting to the formatting reward.

**Intervention (large-to-small group size schedule)**    To stabilize training, we warm-start training with a larger group size to encourage exploration of different patterns [7] and reduce the group size to the default value of $G = 4$ whenever the models make the phase transition and exit the shortcut.

## C.2    EFFECT OF STANDARD DEVIATION NORMALIZATION

For our main experiments, we do not apply std normalization to the group advantage, following Liu et al. (2025). Here, we present the results when we do normalize the advantages by the standard deviation in each group (Shao et al., 2024). All results are from the 1.5B model.

### C.2.1    NEGATIVE GRADIENTS INDUCE LONGER RESPONSES

When initializing training with the normalization, the response length increases faster and stays higher, compared to when training without normalization (Figure 13). We observe a similar trend with a different model (Appendix D).

To identify the cause of the phenomenon, we separately train with positive gradients only (set the group advantage $A_i = \max(A_i, 0)$ after the standard deviation normalization) and negative gradients only ($A_i = \min(A_i, 0)$). The response length of the positive-only training run follows the trend of training without the normalization, whereas negative-only training mimics the trend of training with the normalization. This shows that the negative gradients are the sole cause behind the initial increase in the response length.

This behavior can be understood as the effect of the negative gradients on the token probabilities. Initially, weak models do not collect the final output reward, and the possible rewards are within $\{0, 0.1\}$. However, when the rewards are normalized by the standard deviation, the absolute value of the advantage becomes at most 20 times larger. Updates using the negative gradients greatly reduces the

---

[7]We find that $G = 16$ for the 1.5B and $G = 8$ for the 3B model are sufficient to induce the phase transition.

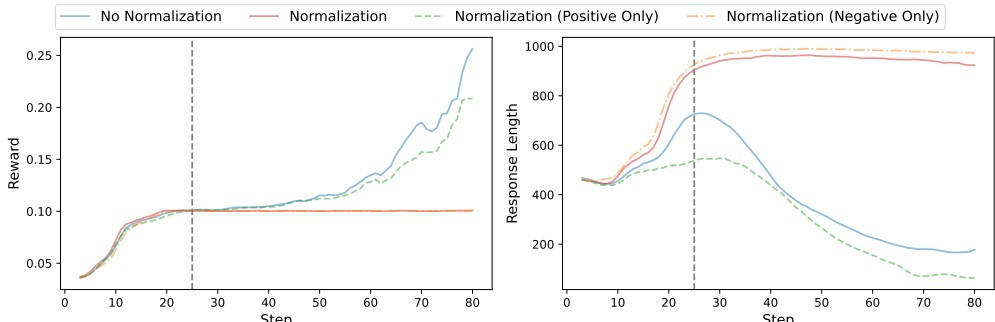

Figure 13: **When normalizing group rewards by standard deviation, negative gradients make responses longer:** **(Left)** training reward; **(Right)** training response length. In the first 25 steps, training reward increases to 0.1 at the same pace across all runs. However, with standard deviation normalization (red), the response length peaks high, and the training reward stalls. Training with only positive gradients (green dotted) and negative gradients (yellow dotted) respectively mimic the behavior of training without (blue) and with (red) normalization.

probability of all tokens generated, and in particular, the generation of the EOS character (which usually occurs after the `<answer></answer>` tags). After the probability of generating EOS token is lowered sufficiently, the model tends to continue generating tokens after the `<answer></answer>` tags.

### C.2.2 STANDARD DEVIATION NORMALIZATION LEADS TO WORSE PERFORMANCE

We additionally notice that normalizing by the standard deviation leads to a worse performance at the end of the training (Table 7).

Table 7: **Effect of standard deviation normalization on final performance.** All-correct@32 (All@), Average Accuracy (Avg), and Pass@32 (Pass@) with or without standard deviation normalization during group rewards computation. Normalizing generally hurts final performance.

|  | No Normalization | | | Normalization | | |
|---|---|---|---|---|---|---|
|  | All@ | Avg | Pass@ | All@ | Avg | Pass@ |
| 1.5B | 0.66 | 0.85 | 0.94 | 0.36 | 0.60 | 0.86 |
| 3B | 0.61 | 0.84 | 0.95 | 0.66 | 0.86 | 0.96 |

### C.3 EFFECT OF NUMBER OF ROLLOUTS

In Appendix C.1, we increase the number of rollouts only during the initial phase of training to induce the phase transition necessary for learning. Here, we explore maintaining the increased number of rollouts ($G = 16$ for the 1.5B model and $G = 8$ for the 3B model) until the end of the training.

We notice that entropy remains slightly higher than training with a small group size ($G = 4$) and the training reward may start decreasing before 1 epoch (around 0.5 epoch for 1.5B model and 0.75 epoch for 3B model). We stop training whenever we observe a decline in the training reward, and evaluate the final performance. The performance of these models is slightly worse than training switching to a smaller rollout mid-training and significantly worse than fully training with $G = 4$ (Table 8). Other than the initial training stability, we observe no benefit of training with a larger group size.

Table 8: **Effect of group size on final performance.** All-correct@32 (All@), Average Accuracy (Avg), and Pass@32 (Pass@) on different choices of group sizes. A larger number of rollouts generally hurts final performance.

| | Small | | | Large | | | Switch | | |
|---|---|---|---|---|---|---|---|---|---|
| | All@ | Avg | Pass@ | All@ | Avg | Pass@ | All@ | Avg | Pass@ |
| 1.5B | 0.66 | 0.85 | 0.94 | 0.46 | 0.78 | 0.93 | 0.46 | 0.79 | 0.95 |
| 3B | 0.61 | 0.84 | 0.95 | 0.46 | 0.78 | 0.94 | 0.48 | 0.83 | 0.98 |

### C.4 Effect of Held-out Patterns

#### C.4.1 Generalization to Held-Out Patterns (More Details)

Here, we continue the discussion from Section 4.4. When we remove the pattern $A/B + C$, we must also remove all patterns which directly include $A/B + C$ as a subpattern, or patterns that can collapse into $A/B + C$, once some subtasks have been solved. See Table 9 for the full list.

Table 9: **The list of patterns removed, when holding out** $A/B + C$**.** We additionally remove 6 patterns that contain $A/B + C$ as a subpattern and 8 more that can be collapsed into $A/B + C$ once a subtask has been solved.

| Pattern | Explanation | Canonical Pattern | Structure |
|---|---|---|---|
| $A/B + C$ | Itself | $A/B + C$ | $[2] + [1]$ |
| $(A/B + C) + D$ | Direct subpattern | $A/B + C + D$ | $[3] + [1]$ |
| $(A/B + C) - D$ | Direct subpattern | $A/B + C - D$ | $[3] - [1]$ |
| $(A/B + C) \times D$ | Direct subpattern | $(A/B + C) \times D$ | $[3] \times [1]$ |
| $(A/B + C)/D$ | Direct subpattern | $(A/B + C)/D$ | $[3] \div [1]$ |
| $A - (B/C + D)$ | Direct subpattern | $A - B/C - D$ | $[1] - [3]$ |
| $A/(B/C + D)$ | Direct subpattern | $A/(B/C + D)$ | $[1] \div [3]$ |
| $A/B + (C \times D)$ | Combine $C \times D$ | $A \times B + C/D$ | $[2] + [2]$ |
| $A/B + (C/D)$ | Combine $C/D$ | $A/B + C/D$ | $[2] + [2]$ |
| $A/(B + C) + D$ | Combine $B + C$ | $A/(B + C) + D$ | $[3] + [1]$ |
| $A/(B - C) + D$ | Combine $B - C$ | $A/(B - C) + D$ | $[3] + [1]$ |
| $(A + B)/C + D$ | Combine $A + B$ | $(A + B)/C + D$ | $[3] + [1]$ |
| $(A - B)/C + D$ | Combine $A - B$ | $(A - B)/C + D$ | $[3] + [1]$ |
| $(A \times B)/C + D$ | Combine $A \times B$ | $A \times B/C + D$ | $[3] + [1]$ |
| $(A/B)/C + D$ | Combine $A/B$ | $A/B/C + D$ | $[3] + [1]$ |

#### C.4.2 What Patterns are Necessary for the Model to Learn?

We further explore the effect of heldout patterns during training. We train the `Qwen2.5-1.5B` model with the following heldout strategies:

- Only 3 / Only 4: Train only with patterns from specific puzzle sizes.
- No Hard: Train without patterns of shape $[1] - [3]$ or $[1] \div [3]$ (Section 4.2)
- Random $p$: Train only with randomly selected $p\%$ of the patterns

Following Appendix C.1, we warm-start the training with a larger group size $G = 16$ and check how many runs are successful out of 3; i.e., model escapes the formatting-only shortcut after something "clicks" (Table 10). When hard patterns are not a part of the training ("Only 3" or "No hard"), the model does not have access to learn from high-entropy, high-reward rollouts and fail to learn. However, training only with $n = 4$ patterns still leads to successful training. Additionally, the diversity of the patterns also seems to be helpful for the model — the probability of a successful run increases with the number of unique patterns in the dataset.

Table 10: **Number of successful runs (out of 3) per heldout strategy:** The existence of harder patterns and an overall wide range of patterns seems necessary for the 1.5B model to learn.

|  | Only 3 | Only 4 | No Hard | Random 30 | Random 40 | Random 50 |
|---|---|---|---|---|---|---|
| # Successful Runs | 0 | 3 | 0 | 1 | 2 | 3 |

### C.4.3 MODELS CAN GENERALIZE TO UNSEEN PATTERNS (RANDOM 50)

Even when the model is trained with only a randomly selected 50% of the patterns, more than 95% of the patterns are `present` in the final checkpoint, and the model scores near 70% average accuracy. The 3B model is able to generalize to more patterns, whereas 4% of patterns are `Absent` from the reasoning of the 1.5B model and another 4% are always applied incorrectly (Table 11). Additionally, the 1.5B model is able to generalize to $n = 5$ patterns, but the strength of the generalization is weaker than having been trained on the full set of patterns (Table 1).

Table 11: **Final performance of models trained without random 50% of patterns:** All-correct@32 (All@), Average Accuracy (Avg), and Pass@32 (Pass@) of models. Models can generalize to most unseen patterns. 3B model is able to generalize to more diverse patterns than 1.5B, but has a lower All-correct@32.

|  | $n=4$ | | | | | $n=5$ | | |
|---|---|---|---|---|---|---|---|---|
|  | All@ | Avg | Pass@ | Absent (%) | Incorrect (%) | All@ | Avg | Pass@ |
| 1.5B | 0.47 | 0.68 | 0.82 | 4 | 4 | 0.06 | 0.25 | 0.52 |
| 3B | 0.35 | 0.69 | 0.88 | 1 | 2 | - | - | - |

### C.5 CHOICE OF RL ALGORITHM

Here, we investigate the effect of the RL algorithm. We replace the GRPO training with PPO (Schulman et al., 2017). Due to higher computational constraints, we only train with 1 seed.

1.5B does not escape the formatting-only shortcut as in Appendix C.1. 3B and 7B improve training accuracy faster with respect to the same number of gradient updates. However, training entropy decreases much faster during PPO training, which translates to a lower Pass@32 but a higher All-correct@32 (Table 12). Compared to GRPO, PPO tends to push the model behavior on each pattern to one extreme — there are simultaneously more 1) `Absent` patterns and 2) patterns with precision of 100%.

Table 12: **Effect of RL algorithm on final performance.** All-correct@32 (All@), Average Accuracy (Avg), and Pass@32 (Pass@) of models. Using PPO instead of GRPO improves All-correct@32 at the cost of Pass@32.

|  | GRPO | | | PPO | | |
|---|---|---|---|---|---|---|
|  | All@ | Avg | Pass@ | All@ | Avg | Pass@ |
| 3B | 0.61 | 0.84 | 0.95 | 0.65 | 0.81 | 0.90 |
| 7B | 0.62 | 0.83 | 0.94 | 0.76 | 0.85 | 0.93 |

## D  RESULTS ON A DIFFERENT MODEL FAMILY

We additionally conduct some ablations studies on the `Llama-3.2` models (1B and 3B) and a `Llama-3.1` model (8B). We take the chat template from the `Instruct` variant but discard most of the system prompt, since the original version is too verbose and is out-of-distribution for the base model. See Appendix E for the exact chat template. We apply the same set of training hyperparmaters.

### D.1  FORMATTING-ONLY SHORTCUT

Across all model sizes (even the 8B model), `Llama-3.2` and `Llama-3.1` models also find a similar formatting-only shortcut as outlined in Appendix C.1. `Llama-3.1-8B` is able to escape the shortcut without a manual intervention ($G = 4$ rollouts throughout training), but the `Llama-3.2` models are unable to escape the shortcut. By applying the group size switching strategy ($G = 16$ for `Llama-3.2-1B` and $G = 8$ for `Llama-3.2-3B`), the models are able to escape the shortcut on 2 out of 3 runs each. This shows that the mitigation strategy for the shortcut is applicable to another model family, but the strength of its effect may depend on the performance of the base model.

### D.2  LLAMA MODELS DO NOT PROPERLY LEARN

Even after escaping the formatting-only shortcut, the `Llama` models do not properly learn to use chain-of-thought to improve its answers. Instead, the models generally guess one expression and pad it with repeated answers or gibberish text.

### D.3  AFTER SFT ON BASIC FORMATTING, RL SHOWS THE SAME TREND

We speculate that the `Llama` base models may require a little more training on the correct formatting (including the `<think>` and `<answer>` tags). We SFT the `Llama-3.2-3B` model on examples from out dataset to teach the formatting.[8] RL training on top of this warmed-up checkpoint shows a similar learning dynamics as in the main part of the paper (Figure 15).

### D.4  EFFECT OF STANDARD DEVIATION NORMALIZATION

We train with standard deviation normalization and reproduce its effect on the response length as outlined in Appendix C.2 and report the result in Figure 14.

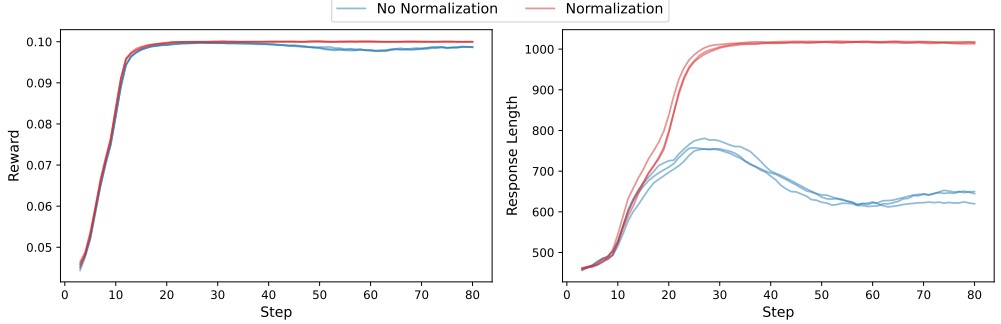

Figure 14: **Effect of standard deviation normalization on response length:** We reproduce the results from Figure 13.

---

[8]We apply the random CoT template and the hyperparameters in Section 4.5. We manually stop training after 50 gradient updates (before the learning rate reaches max value) when the training loss decreases below 1.

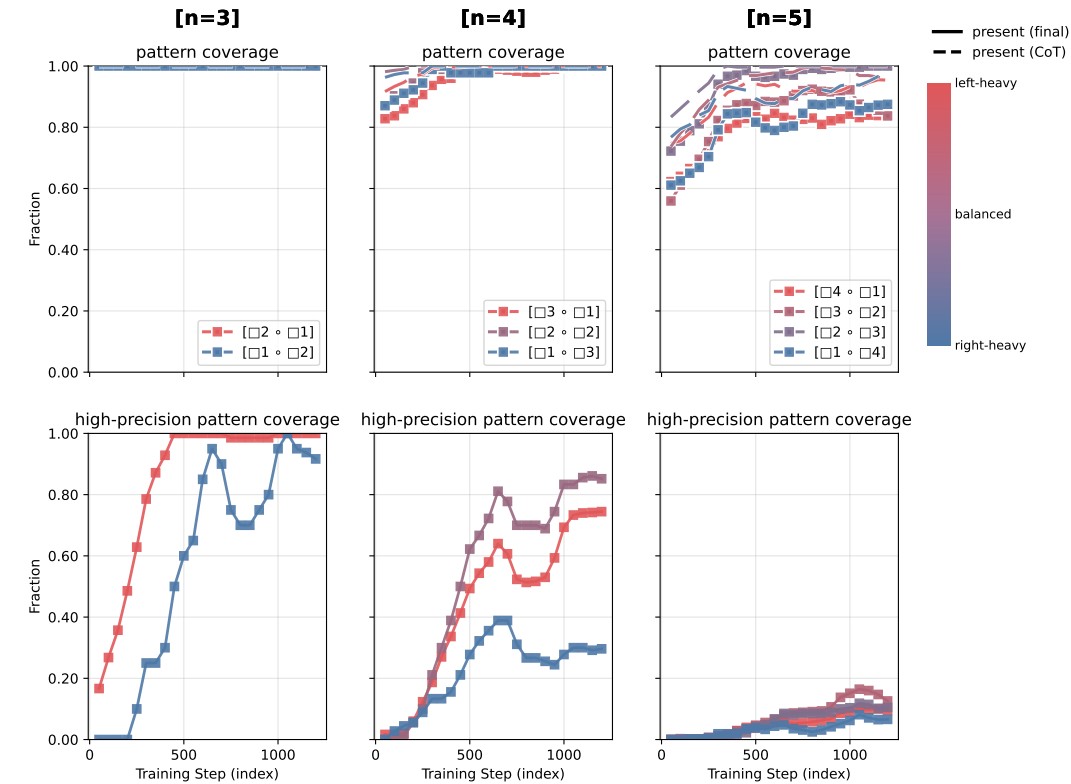

Figure 15: **Learning dynamics at a glance (Llama-3.2-3B after lightweight SFT)**. The same conclusion from Figure 4 holds about the hierarchy of difficulty by subtree structure: balanced < left-heavy < right-heavy.

# E PROMPTS USED FOR TRAINING

---

**Prompt Template for RL**

A conversation between User and Assistant. The user asks a question, and the Assistant solves it. The assistant first thinks about the reasoning process in the mind and then provides the user with the answer.
User: Using the numbers {numbers}, create an equation that equals {target}. You can use basic arithmetic operations (+, -, *, /) and each number can only be used once. Show your work in <think> </think> tags. And return the final answer in <answer> </answer> tags, for example <answer> (1 + 2) / 3 </answer>.
Assistant: Let me solve this step by step.
<think>

---

**Prompt Template for SFT**

A conversation between User and Assistant. The user asks a question, and the Assistant solves it. The assistant first thinks about the reasoning process in the mind and then provides the user with the answer.
User: Using the numbers {numbers}, create an equation that equals {target}. You can use basic arithmetic operations (+, -, *, /) and each number can only be used once. Show your work in <think> </think> tags. And return the final answer in <answer> </answer> tags, for example <answer> (1 + 2) / 3 </answer>.
Assistant: Let me solve this step by step.
<think>Using the numbers {numbers}, create an equation that equals {target}.{CoT text}</think>
<answer>{Solution text}</answer>

---

**Example CoT Text for SFT**

8 / 8 + 3 / 3 = 1 (Incorrect). 3 + 8 + 8 - 3 = 16 (Incorrect). 3 * 3 / 8 - 8 = -6.87 (Incorrect). 8 / (3 - 8 / 3) = 24 (Correct).

---

**Example Solution Text for SFT**

8 / (3 - 8 / 3)

---

**Chat Template for `Llama` Models**

<|begin_of_text|><|start_header_id|>user<|end_header_id|>

{User prompt}<|eot_id|><|start_header_id|>assistant<|end_header_id|>

# F  ALTERNATE WAYS TO CANONICALIZE PATTERNS

A central claim of this work is that the compositional structure of a problem solution (specifically balanced vs. unbalanced) dictates its difficulty for the model. This analysis, however, relies on our specific method for mapping syntactically diverse expressions to unique canonical operators. A valid concern is whether our findings are an artifact of this mapping. For instance, our normalization rules (Appendix A.1) resolve ambiguities by systematically preferring left-heavy structures (e.g., mapping both $9 \times (3 + 2)$ and $(3 + 2) \times 9$ to the canonical form $(A + B) \times C$). This choice could interact with an intrinsic bias of the model towards left-to-right sequential generation, thereby confounding the analysis of structural difficulty. To isolate the effect of structure from our analytical choices, we conduct two ablations to test the robustness of our conclusions.

**Reversed Canonicalization Preference**  First, we investigate whether our findings hold under an alternative normalization scheme. We invert the rule that resolves ambiguity based on subtree weight, now enforcing a preference for *right-heavy* structures. For example, under this new canonicalization scheme, both $9 \times (3 + 2)$ and $(3 + 2) \times 9$ to the canonical form $A \times (B + C)$. We then redo the analysis. The results, shown in Figure 16, indicate that our central observations are unaffected by this change. We observe the same hierarchy of difficulty as reported in Section 4.2: balanced patterns are mastered most reliably, while the structures that were originally classified as right-heavy (and additionally structures now included in the right-heavy grouping) remain the most challenging. This suggests that the difficulty arises from the procedural requirement to commit to an operator before a complex subroutine is generated, rather than an arbitrary choice of a left- or right-associative canonical representation.

**Analysis Without Canonicalization**  Second, we remove the canonicalization process entirely and analyze the raw expressions generated by the model. In this setting, a single problem may have multiple correct solutions corresponding to different tree shapes (e,g, $9 \times (3 + 2)$ would map to $A \times (B + C)$ and $(3 + 2) \times 9$ would map to $(A + B) \times C$). We group all valid expressions generated across the evaluation set by their unnormalized tree shapes (left-heavy, balanced, or right-heavy), and measure the per-shape pattern coverage and high-precision pattern coverage. As shown in Figure 17, the structural difficulty persists. However, a model's intrinsic bias toward generating solutions in a particular syntactic form can artificially deflate the measured performance of any structural group that includes alternative, algebraically equivalent forms. For example, if a model only ever generates structures like $(A + B) + C$, but not $A + (B + C)$, then the coverage for $[1] \circ [2]$ goes down even though the model can correctly compose two additions.

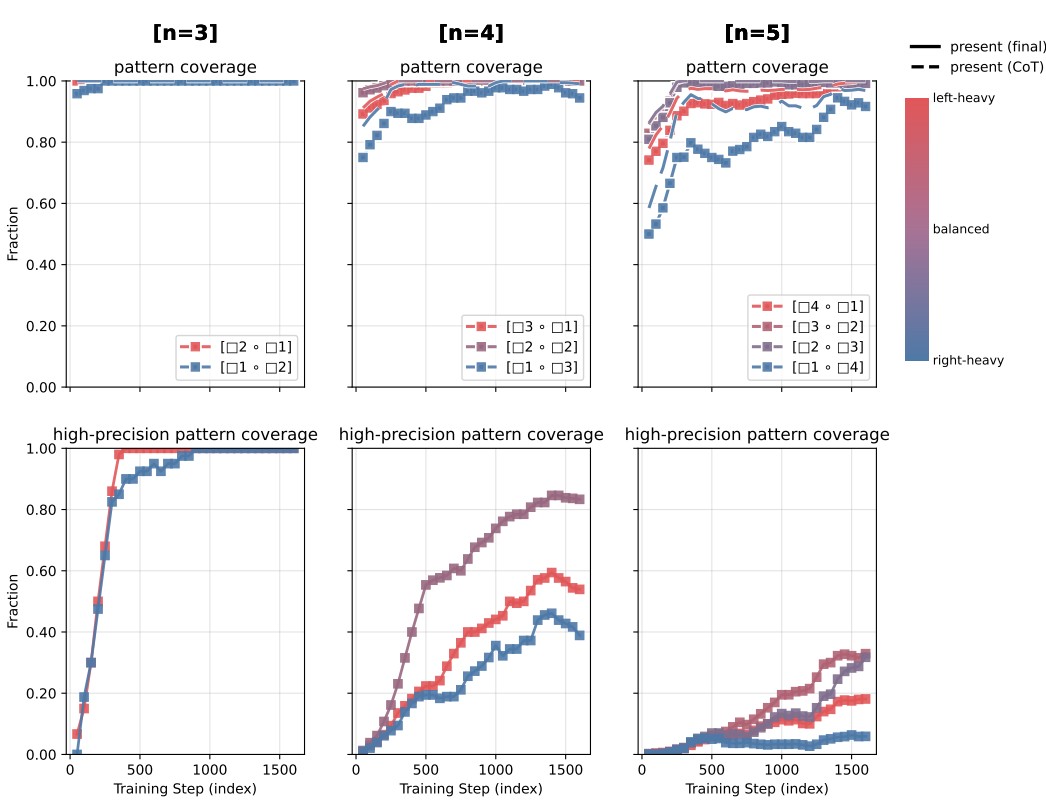

Figure 16: **Learning dynamics at a glance (Qwen2.5-1.5B)** with *reversed* canonicalization. The same conclusion from Figure 4 holds about the hierarchy of difficulty by subtree structure: balanced < left-heavy < right-heavy.

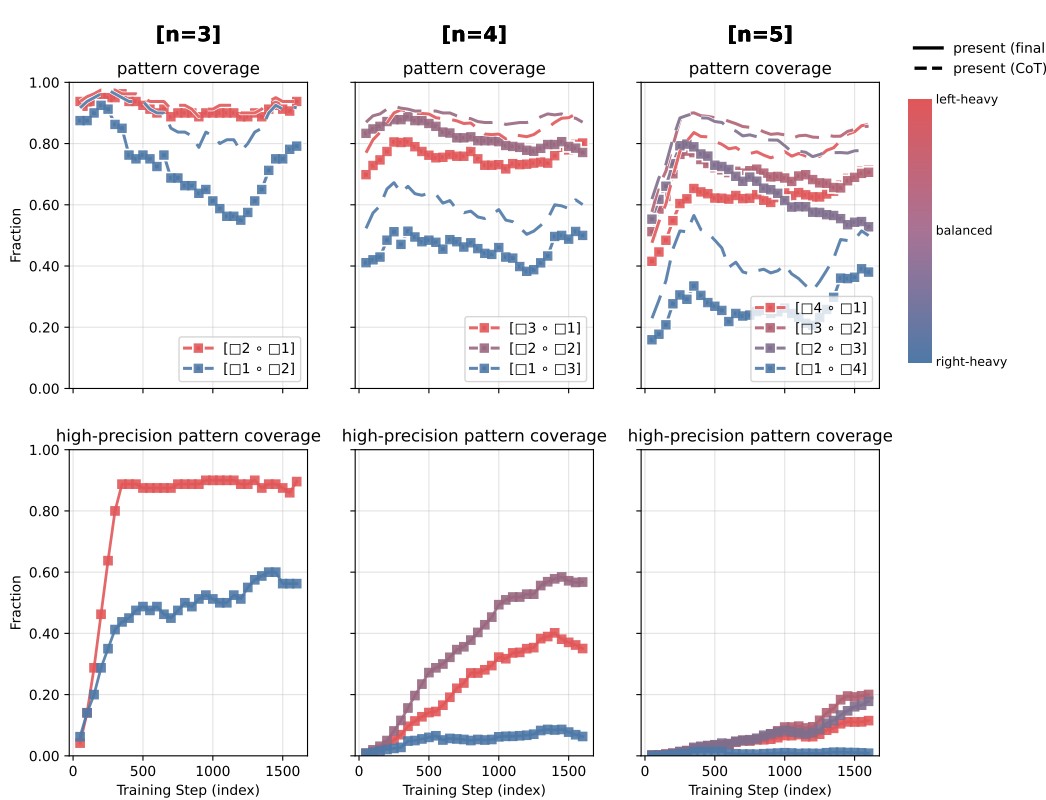

Figure 17: **Learning dynamics at a glance (Qwen2.5-1.5B)** *without* canonicalization. While coverage decreases throughout training (models converge on one of multiple equivalent expressions), the same conclusion from Figure 4 holds about the hierarchy of difficulty by subtree structure: balanced < left-heavy < right-heavy.

# G    ADDITIONAL PLOTS

In Figures 18 to 23, we present the learning dynamics plots for the Qwen2.5-3B/7B models and a different training run for the Qwen2.5-1.5B model (equivalent to Figure 4).

In Figure 26, we present the learning dynamics plots for the Qwen2.5-1.5B model trained on $n \in \{2, 3, 4\}$ (equivalent to Figures 3 and 4). For each of 4 patterns for $n = 2$, we generated 1000 examples for training and 10 examples for testing in the same way as in Appendix A.2.3.

In Figure 27, we present the learning dynamics plots for the Qwen2.5-1.5B model (equivalent to Figure 4) when the high-precision threshold is changed to 70% or 90%.

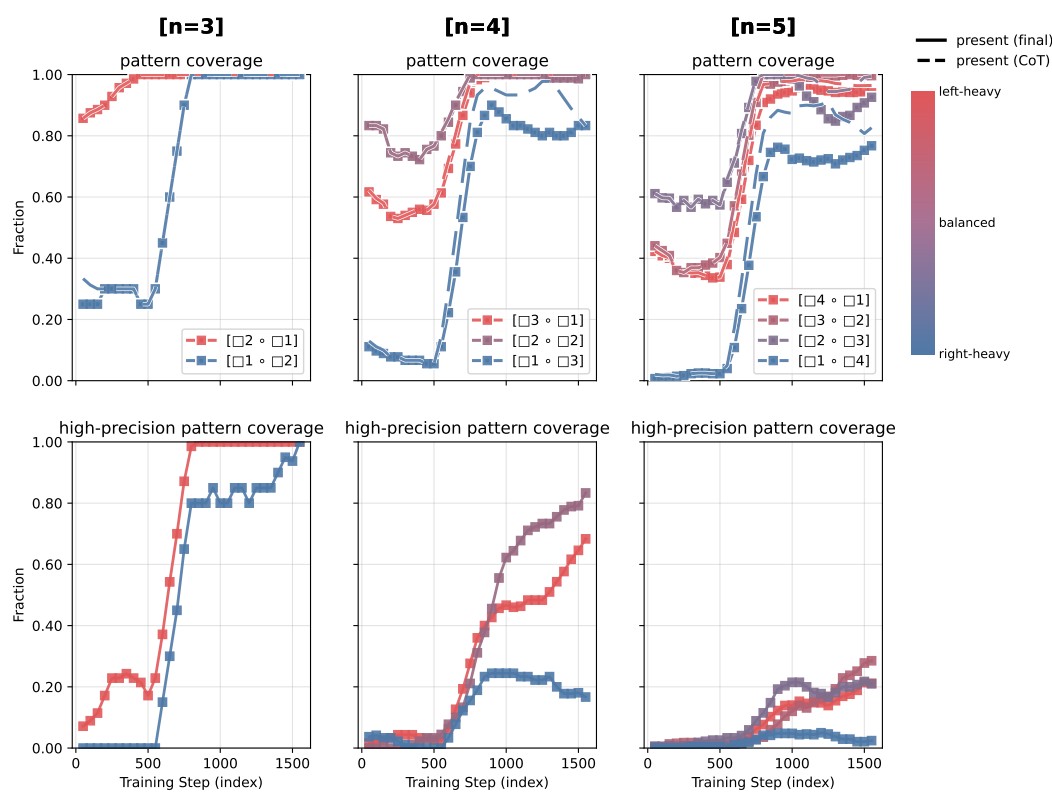

Figure 18: **Learning dynamics at a glance (Qwen2.5-1.5B, another seed)**. The same conclusion from Figure 4 holds about the hierarchy of difficulty by subtree structure: balanced < left-heavy < right-heavy.

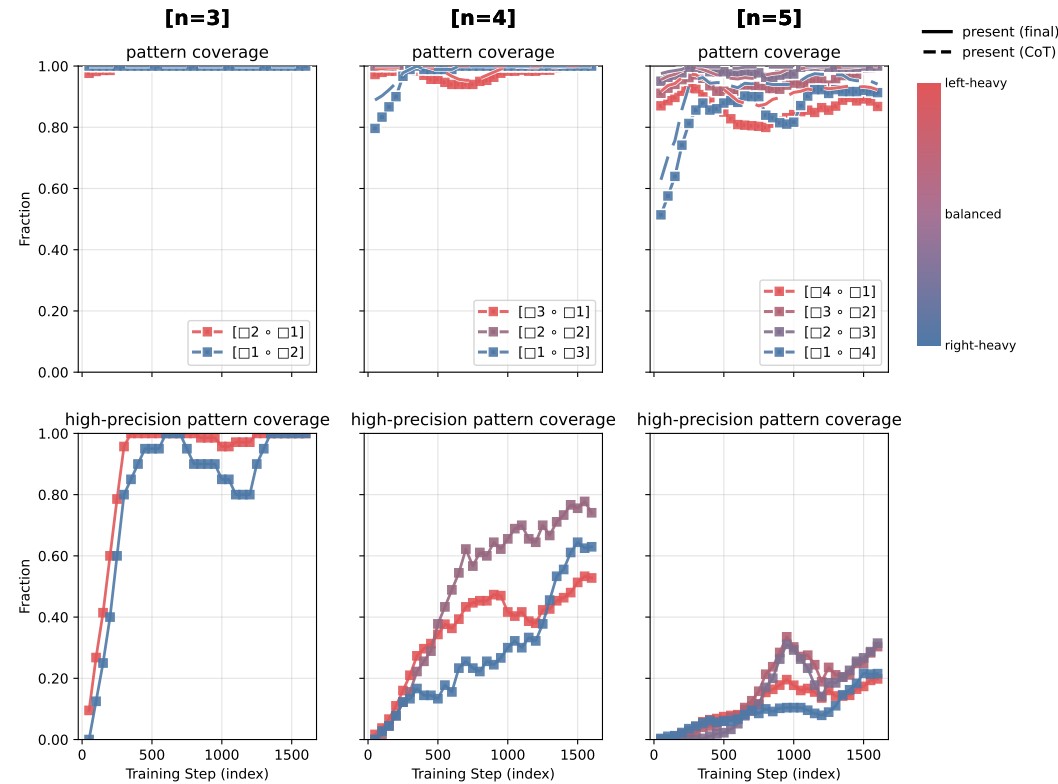

Figure 19: **Learning dynamics at a glance (Qwen2.5-3B)**. The same conclusion from Figure 4 holds about the hierarchy of difficulty by subtree structure: balanced < left-heavy < right-heavy.

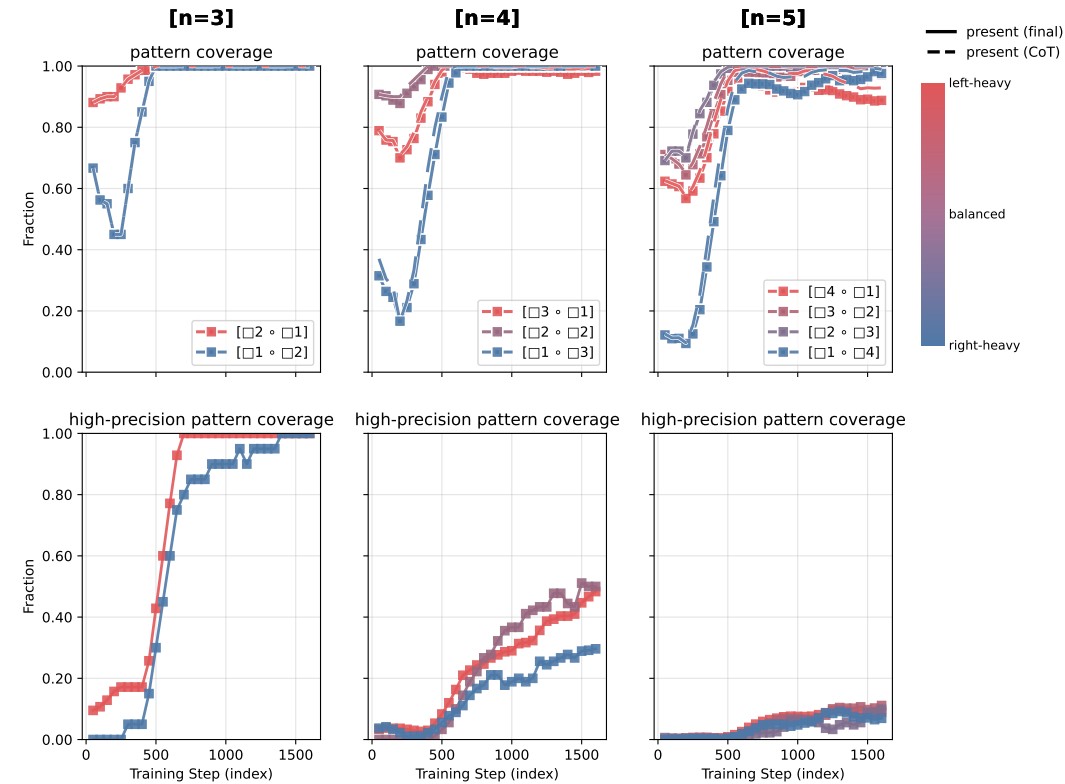

Figure 20: **Learning dynamics at a glance (Qwen2.5-3B, another seed)**. The same conclusion from Figure 4 holds about the hierarchy of difficulty by subtree structure: balanced < left-heavy < right-heavy.

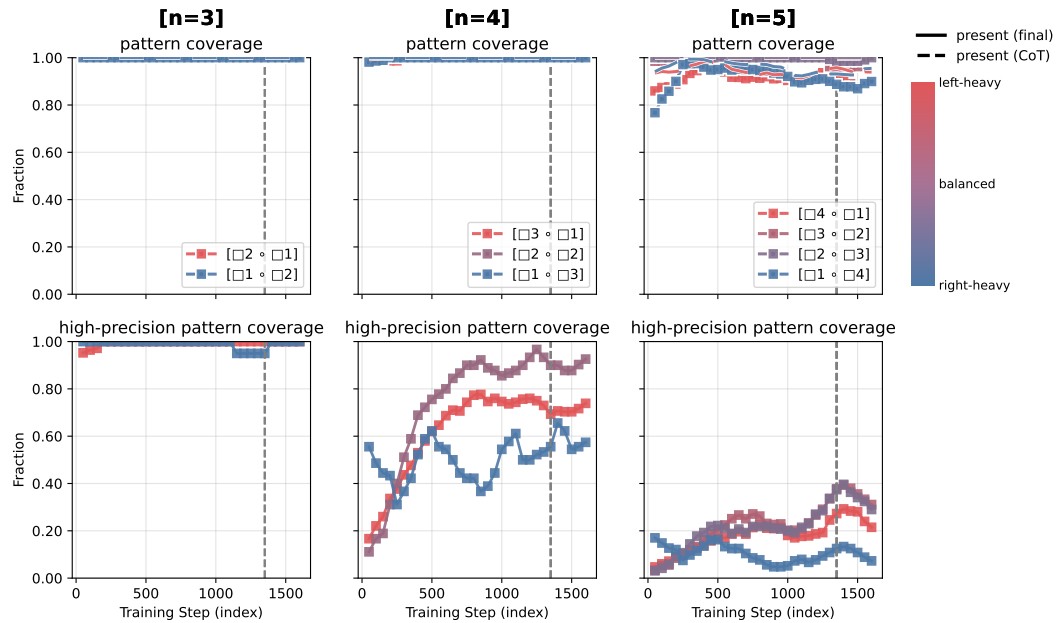

Figure 21: **Learning dynamics at a glance (Qwen2.5-7B)**. The same conclusion from Figure 4 holds about the hierarchy of difficulty by subtree structure: balanced < left-heavy < right-heavy. We take the **best** checkpoint at 1350 steps (grey dotted), after which training reward starts declining.

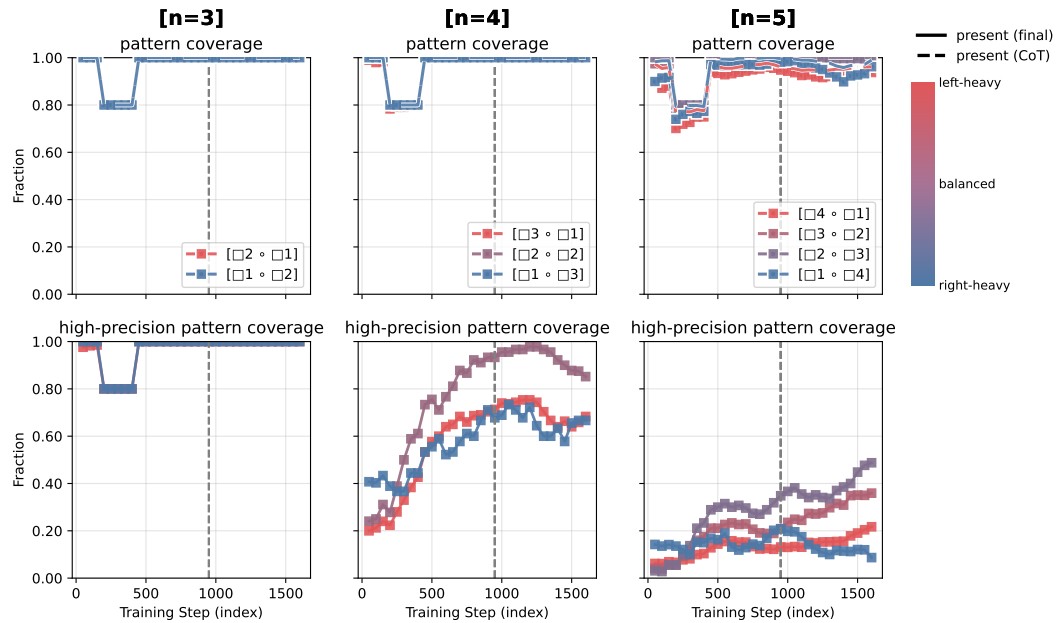

Figure 22: **Learning dynamics at a glance (Qwen2.5-7B, another seed)**. The same conclusion from Figure 4 holds about the hierarchy of difficulty by subtree structure: balanced < left-heavy < right-heavy. We take the **best** checkpoint at 950 steps (grey dotted), after which training reward starts declining.

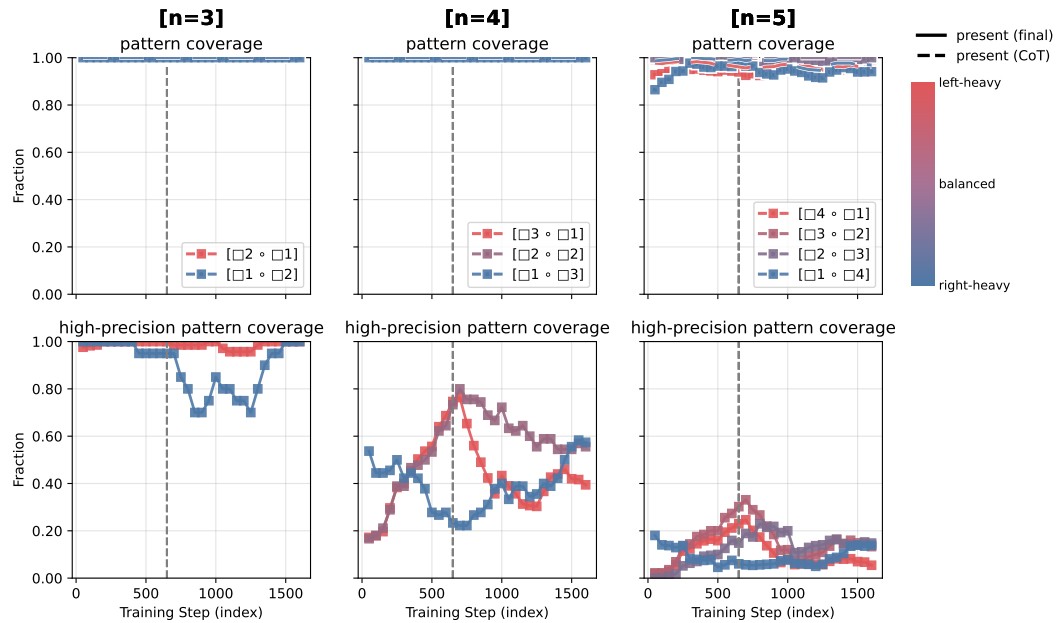

Figure 23: **Learning dynamics at a glance (Qwen2.5-7B, another seed)**. The same conclusion from Figure 4 holds about the hierarchy of difficulty by subtree structure: balanced < left-heavy < right-heavy. We take the **best** checkpoint at 650 steps (grey dotted), after which training reward starts declining.

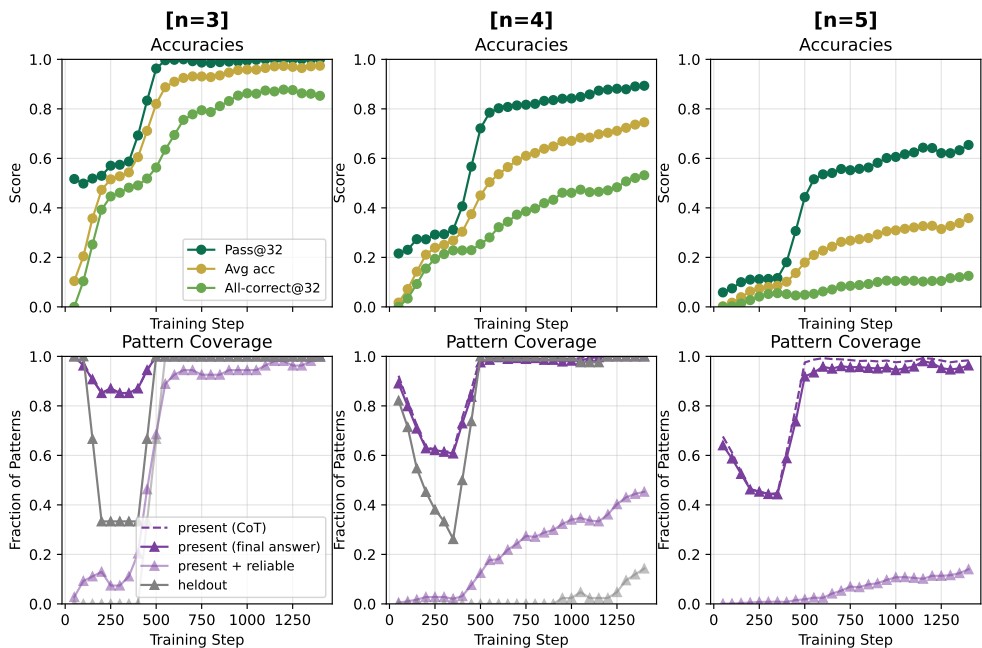

Figure 24: **Generalizes to entire families of held-out compositional patterns (Qwen2.5-1.5B, another seed).** The same conclusion from Figure 5: the model can reuse learned substructures to assemble novel operator-tree shapes it has never been explicitly trained on.

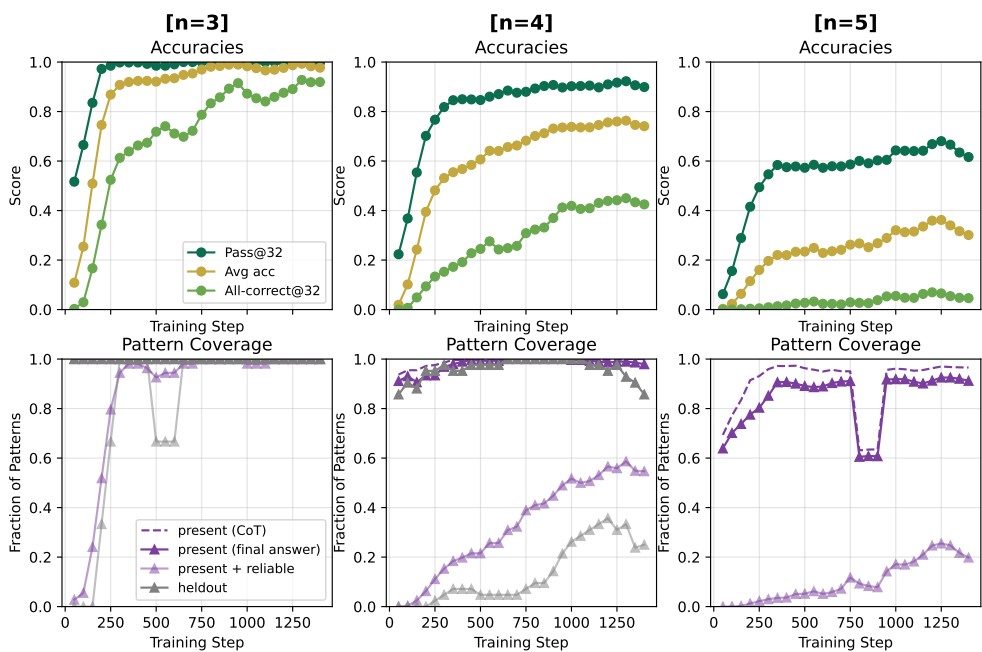

Figure 25: **Generalizes to entire families of held-out compositional patterns (Qwen2.5-1.5B, another seed).** The same conclusion from Figure 5: the model can reuse learned substructures to assemble novel operator-tree shapes it has never been explicitly trained on.

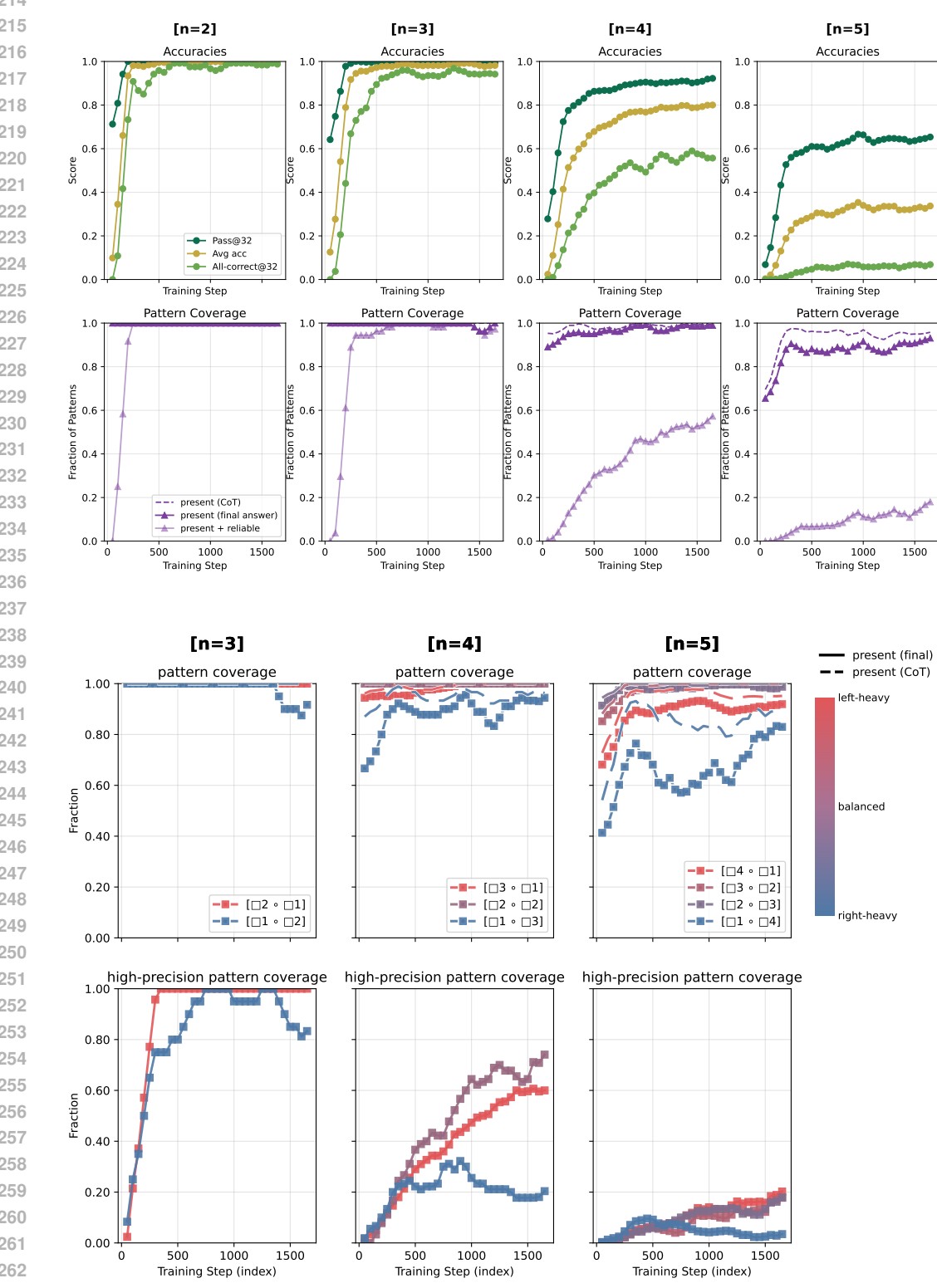

Figure 26: **Learning dynamics at a glance (Qwen2.5-1.5B)** trained on $n \in \{2, 3, 4\}$. The same conclusions from Figures 3 and 4 hold: models exhibit length generalization and master patterns progressively following a hierarchy of difficulty by subtree structure: balanced < left-heavy < right-heavy.

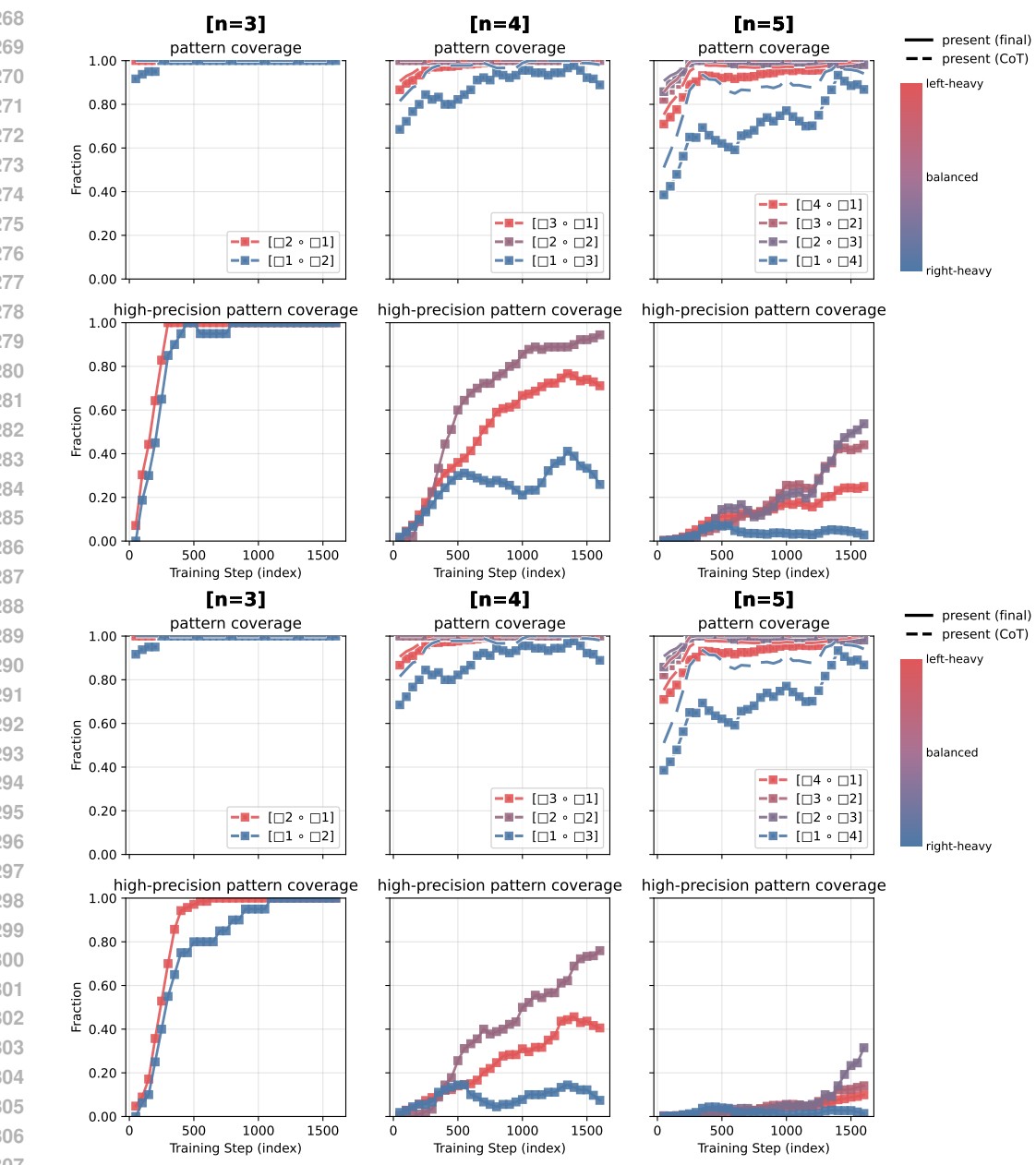

Figure 27: **Learning dynamics at a glance (Qwen2.5-1.5B)** but high-precision defined as precision $\geq 70\%$ (top); $\geq 90\%$ (bottom). The same conclusion from Figure 4 holds about the hierarchy of difficulty by subtree structure: balanced < left-heavy < right-heavy.

## H  LIST OF PATTERNS

Table 13: Expressions and their Canonical Forms (Patterns) (Page 1 of 4)

| Expression | Pattern | Expression | Pattern |
|---|---|---|---|
| $(A + B) + C$ | $A + B + C$ | $A + (B + C)$ | $A + B + C$ |
| $(A + B) - C$ | $A + B - C$ | $A + (B - C)$ | $A + B - C$ |
| $(A + B) \times C$ | $(A + B) \times C$ | $A + (B \times C)$ | $A \times B + C$ |
| $(A + B)/C$ | $(A + B)/C$ | $A + (B/C)$ | $A/B + C$ |
| $(A - B) + C$ | $A + B - C$ | $A - (B + C)$ | $A - B - C$ |
| $(A - B) - C$ | $A - B - C$ | $A - (B - C)$ | $A + B - C$ |
| $(A - B) \times C$ | $(A - B) \times C$ | $A - (B \times C)$ | $A - B \times C$ |
| $(A - B)/C$ | $(A - B)/C$ | $A - (B/C)$ | $A - B/C$ |
| $(A \times B) + C$ | $A \times B + C$ | $A \times (B + C)$ | $(A + B) \times C$ |
| $(A \times B) - C$ | $A \times B - C$ | $A \times (B - C)$ | $(A - B) \times C$ |
| $(A \times B) \times C$ | $A \times B \times C$ | $A \times (B \times C)$ | $A \times B \times C$ |
| $(A \times B)/C$ | $A \times B/C$ | $A \times (B/C)$ | $A \times B/C$ |
| $(A/B) + C$ | $A/B + C$ | $A/(B + C)$ | $A/(B + C)$ |
| $(A/B) - C$ | $A/B - C$ | $A/(B - C)$ | $A/(B - C)$ |
| $(A/B) \times C$ | $A \times B/C$ | $A/(B \times C)$ | $A/B/C$ |
| $(A/B)/C$ | $A/B/C$ | $A/(B/C)$ | $A \times B/C$ |
| $A + (B + (C + D))$ | $A + B + C + D$ | $A + (B + (C - D))$ | $A + B + C - D$ |
| $A + (B + (C \times D))$ | $A \times B + C + D$ | $A + (B + (C/D))$ | $A/B + C + D$ |
| $A + (B - (C + D))$ | $A + B - C - D$ | $A + (B - (C - D))$ | $A + B + C - D$ |
| $A + (B - (C \times D))$ | $A + B - C \times D$ | $A + (B - (C/D))$ | $A + B - C/D$ |
| $A + (B \times (C + D))$ | $(A + B) \times C + D$ | $A + (B \times (C - D))$ | $(A - B) \times C + D$ |
| $A + (B \times (C \times D))$ | $A \times B \times C + D$ | $A + (B \times (C/D))$ | $A \times B/C + D$ |
| $A + (B/(C + D))$ | $A/(B + C) + D$ | $A + (B/(C - D))$ | $A/(B - C) + D$ |
| $A + (B/(C \times D))$ | $A/B/C + D$ | $A + (B/(C/D))$ | $A \times B/C + D$ |
| $A - (B + (C + D))$ | $A - B - C - D$ | $A - (B + (C - D))$ | $A + B - C - D$ |
| $A - (B + (C \times D))$ | $A - B \times C - D$ | $A - (B + (C/D))$ | $A - B/C - D$ |
| $A - (B - (C + D))$ | $A + B + C - D$ | $A - (B - (C - D))$ | $A + B - C - D$ |
| $A - (B - (C \times D))$ | $A \times B + C - D$ | $A - (B - (C/D))$ | $A/B + C - D$ |
| $A - (B \times (C + D))$ | $A - (B + C) \times D$ | $A - (B \times (C - D))$ | $A - (B - C) \times D$ |
| $A - (B \times (C \times D))$ | $A - B \times C \times D$ | $A - (B \times (C/D))$ | $A - B \times C/D$ |
| $A - (B/(C + D))$ | $A - B/(C + D)$ | $A - (B/(C - D))$ | $A - B/(C - D)$ |
| $A - (B/(C \times D))$ | $A - B/C/D$ | $A - (B/(C/D))$ | $A - B \times C/D$ |
| $A \times (B + (C + D))$ | $(A + B + C) \times D$ | $A \times (B + (C - D))$ | $(A + B - C) \times D$ |
| $A \times (B + (C \times D))$ | $(A \times B + C) \times D$ | $A \times (B + (C/D))$ | $(A/B + C) \times D$ |
| $A \times (B - (C + D))$ | $(A - B - C) \times D$ | $A \times (B - (C - D))$ | $(A + B - C) \times D$ |
| $A \times (B - (C \times D))$ | $(A - B \times C) \times D$ | $A \times (B - (C/D))$ | $(A - B/C) \times D$ |
| $A \times (B \times (C + D))$ | $(A + B) \times C \times D$ | $A \times (B \times (C - D))$ | $(A - B) \times C \times D$ |
| $A \times (B \times (C \times D))$ | $A \times B \times C \times D$ | $A \times (B \times (C/D))$ | $A \times B \times C/D$ |
| $A \times (B/(C + D))$ | $A \times B/(C + D)$ | $A \times (B/(C - D))$ | $A \times B/(C - D)$ |
| $A \times (B/(C \times D))$ | $A \times B/C/D$ | $A \times (B/(C/D))$ | $A \times B \times C/D$ |
| $A/(B + (C + D))$ | $A/(B + C + D)$ | $A/(B + (C - D))$ | $A/(B + C - D)$ |
| $A/(B + (C \times D))$ | $A/(B \times C + D)$ | $A/(B + (C/D))$ | $A/(B/C + D)$ |
| $A/(B - (C + D))$ | $A/(B - C - D)$ | $A/(B - (C - D))$ | $A/(B + C - D)$ |
| $A/(B - (C \times D))$ | $A/(B - C \times D)$ | $A/(B - (C/D))$ | $A/(B - C/D)$ |
| $A/(B \times (C + D))$ | $A/(B + C)/D$ | $A/(B \times (C - D))$ | $A/(B - C)/D$ |
| $A/(B \times (C \times D))$ | $A/B/C/D$ | $A/(B \times (C/D))$ | $A \times B/C/D$ |
| $A/(B/(C + D))$ | $(A + B) \times C/D$ | $A/(B/(C - D))$ | $(A - B) \times C/D$ |
| $A/(B/(C \times D))$ | $A \times B \times C/D$ | $A/(B/(C/D))$ | $A \times B/C/D$ |
| $A + ((B + C) + D)$ | $A + B + C + D$ | $A + ((B - C) + D)$ | $A + B + C - D$ |
| $A + ((B \times C) + D)$ | $A \times B + C + D$ | $A + ((B/C) + D)$ | $A/B + C + D$ |

Table 14: Expressions and their Canonical Forms (Patterns) (Page 2 of 4)

| Expression | Pattern | Expression | Pattern |
|---|---|---|---|
| $A + ((B + C) - D)$ | $A + B + C - D$ | $A + ((B - C) - D)$ | $A + B - C - D$ |
| $A + ((B \times C) - D)$ | $A \times B + C - D$ | $A + ((B/C) - D)$ | $A/B + C - D$ |
| $A + ((B + C) \times D)$ | $(A + B) \times C + D$ | $A + ((B - C) \times D)$ | $(A - B) \times C + D$ |
| $A + ((B \times C) \times D)$ | $A \times B \times C + D$ | $A + ((B/C) \times D)$ | $A \times B/C + D$ |
| $A + ((B + C)/D)$ | $(A + B)/C + D$ | $A + ((B - C)/D)$ | $(A - B)/C + D$ |
| $A + ((B \times C)/D)$ | $A \times B/C + D$ | $A + ((B/C)/D)$ | $A/B/C + D$ |
| $A - ((B + C) + D)$ | $A - B - C - D$ | $A - ((B - C) + D)$ | $A + B - C - D$ |
| $A - ((B \times C) + D)$ | $A - B \times C - D$ | $A - ((B/C) + D)$ | $A - B/C - D$ |
| $A - ((B + C) - D)$ | $A + B - C - D$ | $A - ((B - C) - D)$ | $A + B + C - D$ |
| $A - ((B \times C) - D)$ | $A + B - C \times D$ | $A - ((B/C) - D)$ | $A + B - C/D$ |
| $A - ((B + C) \times D)$ | $A - (B + C) \times D$ | $A - ((B - C) \times D)$ | $A - (B - C) \times D$ |
| $A - ((B \times C) \times D)$ | $A - B \times C \times D$ | $A - ((B/C) \times D)$ | $A - B \times C/D$ |
| $A - ((B + C)/D)$ | $A - (B + C)/D$ | $A - ((B - C)/D)$ | $A - (B - C)/D$ |
| $A - ((B \times C)/D)$ | $A - B \times C/D$ | $A - ((B/C)/D)$ | $A - B/C/D$ |
| $A \times ((B + C) + D)$ | $(A + B + C) \times D$ | $A \times ((B - C) + D)$ | $(A + B - C) \times D$ |
| $A \times ((B \times C) + D)$ | $(A \times B + C) \times D$ | $A \times ((B/C) + D)$ | $(A/B + C) \times D$ |
| $A \times ((B + C) - D)$ | $(A + B - C) \times D$ | $A \times ((B - C) - D)$ | $(A - B - C) \times D$ |
| $A \times ((B \times C) - D)$ | $(A \times B - C) \times D$ | $A \times ((B/C) - D)$ | $(A/B - C) \times D$ |
| $A \times ((B + C) \times D)$ | $(A + B) \times C \times D$ | $A \times ((B - C) \times D)$ | $(A - B) \times C \times D$ |
| $A \times ((B \times C) \times D)$ | $A \times B \times C \times D$ | $A \times ((B/C) \times D)$ | $A \times B \times C/D$ |
| $A \times ((B + C)/D)$ | $(A + B) \times C/D$ | $A \times ((B - C)/D)$ | $(A - B) \times C/D$ |
| $A \times ((B \times C)/D)$ | $A \times B \times C/D$ | $A \times ((B/C)/D)$ | $A \times B/C/D$ |
| $A/((B + C) + D)$ | $A/(B + C + D)$ | $A/((B - C) + D)$ | $A/(B + C - D)$ |
| $A/((B \times C) + D)$ | $A/(B \times C + D)$ | $A/((B/C) + D)$ | $A/(B/C + D)$ |
| $A/((B + C) - D)$ | $A/(B + C - D)$ | $A/((B - C) - D)$ | $A/(B - C - D)$ |
| $A/((B \times C) - D)$ | $A/(B \times C - D)$ | $A/((B/C) - D)$ | $A/(B/C - D)$ |
| $A/((B + C) \times D)$ | $A/(B + C)/D$ | $A/((B - C) \times D)$ | $A/(B - C)/D$ |
| $A/((B \times C) \times D)$ | $A/B/C/D$ | $A/((B/C) \times D)$ | $A \times B/C/D$ |
| $A/((B + C)/D)$ | $A \times B/(C + D)$ | $A/((B - C)/D)$ | $A \times B/(C - D)$ |
| $A/((B \times C)/D)$ | $A \times B/C/D$ | $A/((B/C)/D)$ | $A \times B \times C/D$ |
| $(A + B) + (C + D)$ | $A + B + C + D$ | $(A + B) + (C - D)$ | $A + B + C - D$ |
| $(A + B) + (C \times D)$ | $A \times B + C + D$ | $(A + B) + (C/D)$ | $A/B + C + D$ |
| $(A - B) + (C + D)$ | $A + B + C - D$ | $(A - B) + (C - D)$ | $A + B - C - D$ |
| $(A - B) + (C \times D)$ | $A \times B + C - D$ | $(A - B) + (C/D)$ | $A/B + C - D$ |
| $(A \times B) + (C + D)$ | $A \times B + C + D$ | $(A \times B) + (C - D)$ | $A \times B + C - D$ |
| $(A \times B) + (C \times D)$ | $A \times B + C \times D$ | $(A \times B) + (C/D)$ | $A \times B + C/D$ |
| $(A/B) + (C + D)$ | $A/B + C + D$ | $(A/B) + (C - D)$ | $A/B + C - D$ |
| $(A/B) + (C \times D)$ | $A \times B + C/D$ | $(A/B) + (C/D)$ | $A/B + C/D$ |
| $(A + B) - (C + D)$ | $A + B - C - D$ | $(A + B) - (C - D)$ | $A + B + C - D$ |
| $(A + B) - (C \times D)$ | $A + B - C \times D$ | $(A + B) - (C/D)$ | $A + B - C/D$ |
| $(A - B) - (C + D)$ | $A - B - C - D$ | $(A - B) - (C - D)$ | $A + B - C - D$ |
| $(A - B) - (C \times D)$ | $A - B \times C - D$ | $(A - B) - (C/D)$ | $A - B/C - D$ |
| $(A \times B) - (C + D)$ | $A \times B - C - D$ | $(A \times B) - (C - D)$ | $A \times B + C - D$ |
| $(A \times B) - (C \times D)$ | $A \times B - C \times D$ | $(A \times B) - (C/D)$ | $A \times B - C/D$ |
| $(A/B) - (C + D)$ | $A/B - C - D$ | $(A/B) - (C - D)$ | $A/B + C - D$ |
| $(A/B) - (C \times D)$ | $A/B - C \times D$ | $(A/B) - (C/D)$ | $A/B - C/D$ |
| $(A + B) \times (C + D)$ | $(A + B) \times (C + D)$ | $(A + B) \times (C - D)$ | $(A + B) \times (C - D)$ |
| $(A + B) \times (C \times D)$ | $(A + B) \times C \times D$ | $(A + B) \times (C/D)$ | $(A + B) \times C/D$ |
| $(A - B) \times (C + D)$ | $(A + B) \times (C - D)$ | $(A - B) \times (C - D)$ | $(A - B) \times (C - D)$ |
| $(A - B) \times (C \times D)$ | $(A - B) \times C \times D$ | $(A - B) \times (C/D)$ | $(A - B) \times C/D$ |

Table 15: Expressions and their Canonical Forms (Patterns) (Page 3 of 4)

| Expression | Pattern | Expression | Pattern |
|---|---|---|---|
| $(A \times B) \times (C + D)$ | $(A + B) \times C \times D$ | $(A \times B) \times (C - D)$ | $(A - B) \times C \times D$ |
| $(A \times B) \times (C \times D)$ | $A \times B \times C \times D$ | $(A \times B) \times (C/D)$ | $A \times B \times C/D$ |
| $(A/B) \times (C + D)$ | $(A + B) \times C/D$ | $(A/B) \times (C - D)$ | $(A - B) \times C/D$ |
| $(A/B) \times (C \times D)$ | $A \times B \times C/D$ | $(A/B) \times (C/D)$ | $A \times B/C/D$ |
| $(A + B)/(C + D)$ | $(A + B)/(C + D)$ | $(A + B)/(C - D)$ | $(A + B)/(C - D)$ |
| $(A + B)/(C \times D)$ | $(A + B)/C/D$ | $(A + B)/(C/D)$ | $(A + B) \times C/D$ |
| $(A - B)/(C + D)$ | $(A - B)/(C + D)$ | $(A - B)/(C - D)$ | $(A - B)/(C - D)$ |
| $(A - B)/(C \times D)$ | $(A - B)/C/D$ | $(A - B)/(C/D)$ | $(A - B) \times C/D$ |
| $(A \times B)/(C + D)$ | $A \times B/(C + D)$ | $(A \times B)/(C - D)$ | $A \times B/(C - D)$ |
| $(A \times B)/(C \times D)$ | $A \times B/C/D$ | $(A \times B)/(C/D)$ | $A \times B \times C/D$ |
| $(A/B)/(C + D)$ | $A/(B + C)/D$ | $(A/B)/(C - D)$ | $A/(B - C)/D$ |
| $(A/B)/(C \times D)$ | $A/B/C/D$ | $(A/B)/(C/D)$ | $A \times B/C/D$ |
| $(A + (B + C)) + D$ | $A + B + C + D$ | $(A + (B - C)) + D$ | $A + B + C - D$ |
| $(A + (B \times C)) + D$ | $A \times B + C + D$ | $(A + (B/C)) + D$ | $A/B + C + D$ |
| $(A - (B + C)) + D$ | $A + B - C - D$ | $(A - (B - C)) + D$ | $A + B + C - D$ |
| $(A - (B \times C)) + D$ | $A + B - C \times D$ | $(A - (B/C)) + D$ | $A + B - C/D$ |
| $(A \times (B + C)) + D$ | $(A + B) \times C + D$ | $(A \times (B - C)) + D$ | $(A - B) \times C + D$ |
| $(A \times (B \times C)) + D$ | $A \times B \times C + D$ | $(A \times (B/C)) + D$ | $A \times B/C + D$ |
| $(A/(B + C)) + D$ | $A/(B + C) + D$ | $(A/(B - C)) + D$ | $A/(B - C) + D$ |
| $(A/(B \times C)) + D$ | $A/B/C + D$ | $(A/(B/C)) + D$ | $A \times B/C + D$ |
| $(A + (B + C)) - D$ | $A + B + C - D$ | $(A + (B - C)) - D$ | $A + B - C - D$ |
| $(A + (B \times C)) - D$ | $A \times B + C - D$ | $(A + (B/C)) - D$ | $A/B + C - D$ |
| $(A - (B + C)) - D$ | $A - B - C - D$ | $(A - (B - C)) - D$ | $A + B - C - D$ |
| $(A - (B \times C)) - D$ | $A - B \times C - D$ | $(A - (B/C)) - D$ | $A - B/C - D$ |
| $(A \times (B + C)) - D$ | $(A + B) \times C - D$ | $(A \times (B - C)) - D$ | $(A - B) \times C - D$ |
| $(A \times (B \times C)) - D$ | $A \times B \times C - D$ | $(A \times (B/C)) - D$ | $A \times B/C - D$ |
| $(A/(B + C)) - D$ | $A/(B + C) - D$ | $(A/(B - C)) - D$ | $A/(B - C) - D$ |
| $(A/(B \times C)) - D$ | $A/B/C - D$ | $(A/(B/C)) - D$ | $A \times B/C - D$ |
| $(A + (B + C)) \times D$ | $(A + B + C) \times D$ | $(A + (B - C)) \times D$ | $(A + B - C) \times D$ |
| $(A + (B \times C)) \times D$ | $(A \times B + C) \times D$ | $(A + (B/C)) \times D$ | $(A/B + C) \times D$ |
| $(A - (B + C)) \times D$ | $(A - B - C) \times D$ | $(A - (B - C)) \times D$ | $(A + B - C) \times D$ |
| $(A - (B \times C)) \times D$ | $(A - B \times C) \times D$ | $(A - (B/C)) \times D$ | $(A - B/C) \times D$ |
| $(A \times (B + C)) \times D$ | $(A + B) \times C \times D$ | $(A \times (B - C)) \times D$ | $(A - B) \times C \times D$ |
| $(A \times (B \times C)) \times D$ | $A \times B \times C \times D$ | $(A \times (B/C)) \times D$ | $A \times B \times C/D$ |
| $(A/(B + C)) \times D$ | $A \times B/(C + D)$ | $(A/(B - C)) \times D$ | $A \times B/(C - D)$ |
| $(A/(B \times C)) \times D$ | $A \times B/C/D$ | $(A/(B/C)) \times D$ | $A \times B \times C/D$ |
| $(A + (B + C))/D$ | $(A + B + C)/D$ | $(A + (B - C))/D$ | $(A + B - C)/D$ |
| $(A + (B \times C))/D$ | $(A \times B + C)/D$ | $(A + (B/C))/D$ | $(A/B + C)/D$ |
| $(A - (B + C))/D$ | $(A - B - C)/D$ | $(A - (B - C))/D$ | $(A + B - C)/D$ |
| $(A - (B \times C))/D$ | $(A - B \times C)/D$ | $(A - (B/C))/D$ | $(A - B/C)/D$ |
| $(A \times (B + C))/D$ | $(A + B) \times C/D$ | $(A \times (B - C))/D$ | $(A - B) \times C/D$ |
| $(A \times (B \times C))/D$ | $A \times B \times C/D$ | $(A \times (B/C))/D$ | $A \times B/C/D$ |
| $(A/(B + C))/D$ | $A/(B + C)/D$ | $(A/(B - C))/D$ | $A/(B - C)/D$ |
| $(A/(B \times C))/D$ | $A/B/C/D$ | $(A/(B/C))/D$ | $A \times B/C/D$ |
| $((A + B) + C) + D$ | $A + B + C + D$ | $((A - B) + C) + D$ | $A + B + C - D$ |
| $((A \times B) + C) + D$ | $A \times B + C + D$ | $((A/B) + C) + D$ | $A/B + C + D$ |
| $((A + B) - C) + D$ | $A + B + C - D$ | $((A - B) - C) + D$ | $A + B - C - D$ |
| $((A \times B) - C) + D$ | $A \times B + C - D$ | $((A/B) - C) + D$ | $A/B + C - D$ |
| $((A + B) \times C) + D$ | $(A + B) \times C + D$ | $((A - B) \times C) + D$ | $(A - B) \times C + D$ |
| $((A \times B) \times C) + D$ | $A \times B \times C + D$ | $((A/B) \times C) + D$ | $A \times B/C + D$ |

Table 16: Expressions and their Canonical Forms (Patterns) (Page 4 of 4)

| Expression | Pattern | Expression | Pattern |
|---|---|---|---|
| $((A+B)/C)+D$ | $(A+B)/C+D$ | $((A-B)/C)+D$ | $(A-B)/C+D$ |
| $((A\times B)/C)+D$ | $A\times B/C+D$ | $((A/B)/C)+D$ | $A/B/C+D$ |
| $((A+B)+C)-D$ | $A+B+C-D$ | $((A-B)+C)-D$ | $A+B-C-D$ |
| $((A\times B)+C)-D$ | $A\times B+C-D$ | $((A/B)+C)-D$ | $A/B+C-D$ |
| $((A+B)-C)-D$ | $A+B-C-D$ | $((A-B)-C)-D$ | $A-B-C-D$ |
| $((A\times B)-C)-D$ | $A\times B-C-D$ | $((A/B)-C)-D$ | $A/B-C-D$ |
| $((A+B)\times C)-D$ | $(A+B)\times C-D$ | $((A-B)\times C)-D$ | $(A-B)\times C-D$ |
| $((A\times B)\times C)-D$ | $A\times B\times C-D$ | $((A/B)\times C)-D$ | $A\times B/C-D$ |
| $((A+B)/C)-D$ | $(A+B)/C-D$ | $((A-B)/C)-D$ | $(A-B)/C-D$ |
| $((A\times B)/C)-D$ | $A\times B/C-D$ | $((A/B)/C)-D$ | $A/B/C-D$ |
| $((A+B)+C)\times D$ | $(A+B+C)\times D$ | $((A-B)+C)\times D$ | $(A+B-C)\times D$ |
| $((A\times B)+C)\times D$ | $(A\times B+C)\times D$ | $((A/B)+C)\times D$ | $(A/B+C)\times D$ |
| $((A+B)-C)\times D$ | $(A+B-C)\times D$ | $((A-B)-C)\times D$ | $(A-B-C)\times D$ |
| $((A\times B)-C)\times D$ | $(A\times B-C)\times D$ | $((A/B)-C)\times D$ | $(A/B-C)\times D$ |
| $((A+B)\times C)\times D$ | $(A+B)\times C\times D$ | $((A-B)\times C)\times D$ | $(A-B)\times C\times D$ |
| $((A\times B)\times C)\times D$ | $A\times B\times C\times D$ | $((A/B)\times C)\times D$ | $A\times B\times C/D$ |
| $((A+B)/C)\times D$ | $(A+B)\times C/D$ | $((A-B)/C)\times D$ | $(A-B)\times C/D$ |
| $((A\times B)/C)\times D$ | $A\times B\times C/D$ | $((A/B)/C)\times D$ | $A\times B/C/D$ |
| $((A+B)+C)/D$ | $(A+B+C)/D$ | $((A-B)+C)/D$ | $(A+B-C)/D$ |
| $((A\times B)+C)/D$ | $(A\times B+C)/D$ | $((A/B)+C)/D$ | $(A/B+C)/D$ |
| $((A+B)-C)/D$ | $(A+B-C)/D$ | $((A-B)-C)/D$ | $(A-B-C)/D$ |
| $((A\times B)-C)/D$ | $(A\times B-C)/D$ | $((A/B)-C)/D$ | $(A/B-C)/D$ |
| $((A+B)\times C)/D$ | $(A+B)\times C/D$ | $((A-B)\times C)/D$ | $(A-B)\times C/D$ |
| $((A\times B)\times C)/D$ | $A\times B\times C/D$ | $((A/B)\times C)/D$ | $A\times B/C/D$ |
| $((A+B)/C)/D$ | $(A+B)/C/D$ | $((A-B)/C)/D$ | $(A-B)/C/D$ |
| $((A\times B)/C)/D$ | $A\times B/C/D$ | $((A/B)/C)/D$ | $A/B/C/D$ |

# I  ACKNOWLEDGING LLM USAGE

At various stages of the project, we received the aid of LLMs. Here, we report all such use cases.

- We received aid on writing the code to automate the process of extracting and canonicalizing an expression. The list of $n = 4$ patterns were initially manually mapped and were used as test cases for the LLM generated code. All code was manually verified.

- We received aid on writing the code to draw plots from existing data that were generated from human-written scripts. All code was manually verified.

- We used an LLM for copy-editing tasks, such as rephrasing sentences to improve clarity, conciseness, and flow, as well as for correcting grammatical errors.

