# OpenReview forum: "How Does RL Induce Skill Composition? A Case Study Using Countdown"
_ICLR.cc/2026/Conference — Submitted to ICLR 2026_

### Official Review · Reviewer_1GMS · 2025-11-01

**Soundness:** 3
**Presentation:** 3
**Contribution:** 3
**Rating:** 4
**Confidence:** 4

**Summary:**

This paper examines how reinforcement learning (RL) fine-tuning on large language models (LLMs) can instill compositional problem-solving skills beyond mere length generalization. Using the COUNTDOWN arithmetic puzzle as a test, the authors represent each solution as a binary expression tree where subtrees correspond to reusable subtasks or “skills.” This framework lets them disentangle two types of generalization: length generalization (solving longer sequences by iterating a known algorithm) versus compositional generalization (synthesizing a novel solution procedure from known components).

The key findings are: (1) Models trained on puzzles with $n=3,4$ numbers can generalize to solve puzzles with $n=5,6$ numbers, indicating they recombine learned smaller-step skills to tackle larger problems. (2) The structural complexity of the solution has a profound effect on learnability: shallow, balanced expression trees are mastered more easily and reliably than deep or unbalanced ones, even when problem length is the same. (3) RL can enable some compositional extrapolation: when an entire family of solution patterns (a particular tree shape at $n=3$) is held out during training, the RL-fine-tuned model is later able to synthesize that unseen pattern and even solve higher-order versions of it, using the familiar atomic skills in a new combination. These results suggest that RL fine-tuning does impart the ability to compose known operations in novel ways, not just to execute a fixed reasoning template for more steps.

**Strengths:**

* The paper addresses an important open question – how does RL improve an LLM’s reasoning abilities – with a very clear focus on compositional generalization. It introduces a well-defined pattern framework to distinguish length vs. compositional generalization formally.

* The authors evaluate generalization along two axes (longer sequences, and novel solution structures) and carefully control for each factor.

* Several interesting findings emerge that deepen our understanding of how RL fine-tuning helps reasoning.

**Weaknesses:**

* Generality to Other Domains: the significance is somewhat tied to this specific testbed; the paper would be stronger if it at least discussed or empirically checked whether similar compositional skill reuse occurs in a different domain or a more natural language reasoning dataset.

* Lack of Baseline Comparisons: The study focuses on RL fine-tuning exclusively, so it is unclear to what extent RL is uniquely responsible for the observed generalization gains. There is little discussion of how a simpler fine-tuning approach would fare. For example, would supervised learning on the same training puzzles (using correct solutions as targets) enable any compositional generalization, or is the reinforcement signal crucial? The paper argues that RL “induces OOD generalization beyond what standard metrics…reveal”, but providing a direct comparison (e.g., a baseline model fine-tuned with supervised learning or guided by chain-of-thought prompting/skill composition based prompting) would substantiate RL’s advantage. Without this, one might wonder if the improvements come partly from exposure to many sampled solutions during RL training rather than the intrinsic benefits of the RL objective. In summary, the experimental evidence for RL’s efficacy would be more convincing if alternative training paradigms were evaluated on the same framework.

* A notable omission is the lack of discussion or citation of some prior works on inducing compositional generalization in language models. For instance, skills-in-context prompting as a strategy to unlock compositional reasoning in LLMs, Learning to Follow Language Instructions with Compositional Policies

* Scope of Skills and Definitions is unclear

**Questions:**

* How confident are you that the hierarchy of difficulty observed (balanced vs. right-heavy trees) will translate to other compositional reasoning tasks?

* Did you consider training the model on the same COUNTDOWN tasks using supervised learning (e.g. cross-entropy on ground-truth solutions or with step-by-step solutions) and comparing its generalization to the RL-fine-tuned model?

* The compositional generalization achieved through RL in your work is reminiscent of results achieved by “skills-in-context” prompting in Chen et al. (2023), although via a very different mechanism. Have you considered combining these ideas? For example, could providing a few compositional exemplars in context plus RL fine-tuning yield even stronger generalization, or help the model leapfrog the hard right-heavy cases? A discussion contrasting learning via rewards versus prompting would be enlightening – e.g., perhaps RL is better at exploration, while prompting provides strong guidance, and the two could be complementary.

---

> ### Author Response · Authors · 2025-11-24
> **Response to Reviewer 1GMS (1/2)**
>
> > Generality to Other Domains: the significance is somewhat tied to this specific testbed; the paper would be stronger if it at least discussed or empirically checked whether similar compositional skill reuse occurs in a different domain or a more natural language reasoning dataset.
>
> We thank the reviewer for this comment. Please see the response to all reviewers **Scope and choice of COUNTDOWN**.
>
> > Lack of Baseline Comparisons: The study focuses on RL fine-tuning exclusively, so it is unclear to what extent RL is uniquely responsible for the observed generalization gains. There is little discussion of how a simpler fine-tuning approach would fare. For example, would supervised learning on the same training puzzles (using correct solutions as targets) enable any compositional generalization, or is the reinforcement signal crucial? The paper argues that RL “induces OOD generalization beyond what standard metrics…reveal”, but providing a direct comparison (e.g., a baseline model fine-tuned with supervised learning or guided by chain-of-thought prompting/skill composition based prompting) would substantiate RL’s advantage. Without this, one might wonder if the improvements come partly from exposure to many sampled solutions during RL training rather than the intrinsic benefits of the RL objective. In summary, the experimental evidence for RL’s efficacy would be more convincing if alternative training paradigms were evaluated on the same framework.
>
>
> We thank the reviewer for the suggestion. We would like to note that two supervised fine-tuning (SFT) baselines are already included in Section 4.5. These SFT baselines do not exhibit the hierarchy of task learnability we see under RL, which suggests that this hierarchy is inherent to RL-style optimization rather than simply a consequence of exposure to the COUNTDOWN training data. Section 4.2 and 4.3 show that, when training LLMs with standard RL methods such as GRPO/PPO, there is a clear hierarchy of learnability across tasks even when problem size (and, in some settings, compositional depth) is held constant.
>
> We would also like to clarify that the goal of this work is not to establish RL’s superiority over alternative methods. Instead, our aim is to use RL post-training to “reveal what is learned, in what order, and where generalization fails” (line 25), rather than to make broad claims about RL’s overall efficacy. Our results in Section 4.4 provide evidence that RL post-training can induce genuine compositional generalization. We do not intend to argue that RL is the only method that can induce skill composition; rather, we seek to distinguish our analysis from prior work that claims “skill composition” but in practice only demonstrates length generalization.
>
>
>
> > A notable omission is the lack of discussion or citation of some prior works on inducing compositional generalization in language models. For instance, skills-in-context prompting as a strategy to unlock compositional reasoning in LLMs, Learning to Follow Language Instructions with Compositional Policies
>
> We thank the reviewer for the suggestion. In the updated manuscript, we have included discussion of additional related works that discuss skill compositions in LLMs. We believe that the novelty of our work still remains unchanged.
>
>
> > Scope of Skills and Definitions is unclear
>
>
> We thank the reviewer for pointing this out and have updated our manuscript (see the updated Section 2.2).
>
>
> > How confident are you that the hierarchy of difficulty observed (balanced vs. right-heavy trees) will translate to other compositional reasoning tasks?
>
> We are not sure, but hypothesize that perhaps balanced tasks may still be easier than unbalanced tasks. This would definitely be an interesting avenue for future work.
>
>
> > Did you consider training the model on the same COUNTDOWN tasks using supervised learning (e.g. cross-entropy on ground-truth solutions or with step-by-step solutions) and comparing its generalization to the RL-fine-tuned model?
>
> We would like to note that two supervised fine-tuning (SFT) baselines are already included in Section 4.5. These SFT baselines do not exhibit the hierarchy of task learnability we see under RL, which suggests that this hierarchy is inherent to RL-style optimization rather than simply a consequence of exposure to the COUNTDOWN training data. Section 4.3 shows that, when training LLMs with standard RL methods such as GRPO/PPO, there is a clear hierarchy of learnability across tasks even when problem size (and, in some settings, compositional depth) is held constant.

---

> ### Author Response · Authors · 2025-11-24
> **Response to Reviewer 1GMS (2/2)**
>
> > The compositional generalization achieved through RL in your work is reminiscent of results achieved by “skills-in-context” prompting in Chen et al. (2023), although via a very different mechanism. Have you considered combining these ideas? For example, could providing a few compositional exemplars in context plus RL fine-tuning yield even stronger generalization, or help the model leapfrog the hard right-heavy cases? A discussion contrasting learning via rewards versus prompting would be enlightening – e.g., perhaps RL is better at exploration, while prompting provides strong guidance, and the two could be complementary.
>
> We thank the reviewer for the suggestion. However, we do not apply such methods because our goal is to study **how skill composition arises during standard RL post-training**, and not to achieve the best possible results.
>
> Because we applied the most standard training setup, we were able to remove any confounding factors and pinpoint the source of the findings (learnability of tasks based on the compositional structure) to the RL post-training. In future works, we hope to study how to address this hierarchy of learnability (including ideas from data curriculum).

---

### Official Review · Reviewer_xutx · 2025-11-01

**Soundness:** 2
**Presentation:** 2
**Contribution:** 2
**Rating:** 2
**Confidence:** 4

**Summary:**

This paper studies the open questions of what RL teaches skill composition and how the structure of the composition affects the skill transfer. It reaches the conclusion that RL-only post-training induces OOD generalization beyond what standard metrics can reveal.

**Strengths:**

The study of RL behavior regarding LLM’s skill composition capability for more sophisticated tasks is interesting research.

**Weaknesses:**

1. The study is limited on the COUNTDOWN task. This task is currently considered relatively easier than other frontier reasoning tasks such as AIME, FrontierMath, HLE, etc. It is still questionable the conclusions observed on COUNTDOWN task still holds true for other more sophisticated tasks.

2. The paper proposes to analyze the underlying computational structure of arithmetic solutions such as canonical patterns of reasoning. The method is vaguely written and reads empirical. It is unclear whether its source code will be released or not. If not, the proposed method would be hard for the community to reproduce.

3. The paper misses to cite a few important related work. For instance:

[1] J. Chen et al. 2024. Skills-in-Context: Unlocking Compositionality in Large Language Models. In Findings of the Association for Computational Linguistics: EMNLP 2024.

**Questions:**

None.

---

> ### Author Response · Authors · 2025-11-24
> **Response to Reviewer xutx**
>
> > The study is limited on the COUNTDOWN task. This task is currently considered relatively easier than other frontier reasoning tasks such as AIME, FrontierMath, HLE, etc. It is still questionable the conclusions observed on COUNTDOWN task still holds true for other more sophisticated tasks.
>
> We thank the reviewer for their comment. Please see the response to all reviewers **Scope and choice of COUNTDOWN**.
>
> We would like to emphasize that the goal of this work is to introduce a controlled synthetic setting where compositional “skills” can be precisely defined and quantified. On natural data (including AIME, FrontierMath, HLE, etc) , compositional complexity is confounded with length, making it impossible to isolate true failure modes; in general, it is much harder to define what skills are/aren’t being used. A synthetic testbed is therefore methodologically essential to control for these confounders and identify drivers of reasoning failure. Thus, the COUNTDOWN task introduces a formal lens through which to analyze the acquisition of compositional skills, and helps identify a gap in the literature: prior works conflate skill composition with length generalization.
>
>
>
> > The paper proposes to analyze the underlying computational structure of arithmetic solutions such as canonical patterns of reasoning. The method is vaguely written and reads empirical. It is unclear whether its source code will be released or not. If not, the proposed method would be hard for the community to reproduce.
>
> Could the reviewer clarify which part they found to be vaguely written? We hope to address any concerns about the clarity/presentation as we update our manuscript. We currently describe the exact procedure of converting an arithmetic expression to its underlying pattern in Appendix A.1 (referenced in line 99). In Tables 11-14 (referenced in Appendix A.1), we provide the exact mapping.
>
> We have updated our submission to include the anonymized version of our codebase as supplementary material. We hope this addresses the reviewer’s concern about reproducibility.
>
> > The paper misses to cite a few important related work. For instance: [1] J. Chen et al. 2024. Skills-in-Context: Unlocking Compositionality in Large Language Models. In Findings of the Association for Computational Linguistics: EMNLP 2024.
>
> We thank the reviewer for the suggestion. In the updated manuscript, we have included discussion of additional related works that discuss skill compositions in LLMs. We believe that the novelty of our work still remains unchanged.

---

> > ### Comment · Reviewer_xutx · 2025-11-24
> >
> > Thanks for the responses! I appreciate the justifications on the scope of COUNTDOWN, and the added analysis on the originally missing related work. However, I would choose to keep my original rating 2 due to the following reasons.
> >
> > 1. I am not convinced on the analysis and conclusion if it is only verified with the synthetic COUNTDOWN task.
> > 2. I am concerned with the incremental novelty comparing to the previous work on the topic of skill composition.
> > 3. The paper is not clearly written and brings in difficulties for the community to re-produce or re-apply to other realistic tasks.

---

### Official Review · Reviewer_fN4B · 2025-11-05

**Soundness:** 2
**Presentation:** 3
**Contribution:** 2
**Rating:** 2
**Confidence:** 4

**Summary:**

This work investigates how RL fine-tuning improves compositional reasoning in large language models (LLMs). Using the COUNTDOWN arithmetic task as a controlled testbed, the authors aim to disentangle length generalisation (applying known policies to longer problems) from compositional generalisation (synthesizing new policies from known subskills). They represent solutions as canonical expression trees to analyse structural learning dynamics. Their results demonstrate that RL-finetuned models can generalize to larger problem sizes (length generalisation) and can synthesize unseen patterns (compositional generalisation). Importantly, they reveal a hierarchy of difficulty: balanced tree structures are easier to learn than right-heavy ones, highlighting a “lookahead bottleneck” as a key limitation in RL-based reasoning (which speaks to the traditional exploration problem).

**Strengths:**

- **Significance and originality**: It is highly relevant to the LLM community, offering a new structural framework to disentangle compositional and length generalisation. This distinction is often conflated in prior works.

- **Empirical depth**: There is a rich set of ablations and diagnostics illustrating how RL finetuning impacts sample-efficiency and generalisation across structural variants of COUNTDOWN tasks.

- **Insightful analysis:** The results in Section 4.2 are particularly interesting, showing that structural pattern complexity plays a significant role in learning difficulty. This is potentially because the balanced and left-heavy patterns allow easier exploration (more ways to solve the task) than the right-heavy ones (due to the bottlenecks). Although the paper does not make this connection. This suggests that RL finetuning remains bounded by classic RL exploration challenges.

**Weaknesses:**

## Major
- **Lack of grounding in RL composition literature:** The paper cites none of the foundational and recent works on skill composition in RL, missing a connection to established definitions and frameworks. For example, the foundational work of Todorov [2009] that demonstrated that RL agents could provably compose learned skills to solve new tasks, which was then extended to Boolean composition by Tasse et al [2020] and more general classes of composition operators by Adamczyk et al. [2023ab]. See Mendez et al [2023] for a survey.

- **Ambiguous RL setup:** The RL formulation (state, action, discounting, etc.) is unspecified. It is unclear whether the actions correspond to full vocab tokens or restricted arithmetic operators, creating methodological ambiguity.

- **Non-standard definitions of skills and skill composition:** In the paper an atomic skill is defined as action, specifically selecting an operator (-,+,x,/). They then define a skill composition as simply being a regular policy (a sequence of actions corresponding to operators and integers).  However, a skill in RL is usually defined as a policy or value function (that corresponds to some well defined task/MDP) and a skill composition is a set of operators applied to a set of skills to generate a new skill.

- **Vague formalism:** The main idea in this paper is to disentangle the effect of length and compositional generalization on model generalization. However these two notions of generalization are not mathematically formalized and lack concrete illustrative examples (the definitions given on page 3 are not precise enough). For example Length generalisation is defined as "the ability to apply a known compositional pattern to problems requiring a greater number of atomic operations", but it is unclear what are examples of such problems using the pattern in Figure 1 as the "known" pattern.

- **Missing dataset and statistical details:** Dataset size (total number of examples), number of examples per subtree shape, training vs heldout splits, etc are not fully reported. It is also unclear if any of the results are statistically significant, since no measure of variance or statistical significance are provided for the highly stochastic RL results. This is especially problematic for non-critic based RL methods like GRPO that rely on Monte Carlo value estimates.

- **Questionable experimental claims:**
  - In Section 4.1, the authors claim "Models trained only on n ∈ {3, 4} problems show substantial accuracy improvements on held-out n = 5 puzzles". Improvements over what? Natural baselines like the pretrained model and SFT models are not included for comparison. The authors also claim "These results confirm that models successfully achieve length generalization". This is unjustified given the drastic accuracy decline as n increases. It is also unjustified given that the heldout tasks (n=5,6) look like they are not constrained to only "known" patterns from the training tasks. This is a problem given that the main point of this experiment was to isolate the amount of length generalisation from compositional generalisation. In general this experiment is not properly controlled.
  - The claims in Section 4.4 rely solely on the heldout curve in Fig 4 (bottom-left and bottom-center). This experiment is not only not averaged across multiple training seeds, but it is also not averaged across different choices of held-out patterns.

## Minor
- Undefined abbreviations (e.g., SFT on first use)
- The accuracy of the greedy policy (i.e. when selecting the highest probability action) is not reported.
- What are those metrics in Fig 2 bottom? We can guess but the paper should precisely define them.
- Table 3 is missing the pretrained model without finetuning.

Overall, this paper makes interesting methodological and empirical contribution to understanding how RL induces compositional reasoning in LLMs, but it would benefit from tighter formal definitions, stronger theoretical grounding, and more controlled experimental validation.

[Todorov 2009] E. Todorov. Compositionality of optimal control laws. In Advances in Neural Information Processing Systems.

[Tasse et al. 2020] G. Nangue Tasse, S. James, and B. Rosman. A Boolean task algebra for reinforcement learning. Advances in Neural Information Processing Systems.

[Adamczyk et al. 2023a] J. Adamczyk, A. Arriojas, S. Tiomkin, and R. V. Kulkarni. Utilizing prior solutions for reward shaping and composition in entropy-regularized reinforcement learning. In Proceedings of the AAAI Conference on Artificial Intelligence.

[Adamczyk et al. 2023b] J. Adamczyk, S. Tiomkin, and R. Kulkarni. Leveraging prior knowledge in reinforcement learning via double-sided bounds on the value function. arXiv preprint arXiv:2302.09676, 2023.

[Mendez et al 2023] J. A. Mendez and E. Eaton. How to Reuse and Compose Knowledge for a Lifetime of Tasks: A Survey on Continual Learning and Functional Composition. TMLR.

**Questions:**

1. Why depart from the standard RL definitions of “skills” and “skill composition”? How would the results or interpretations change under conventional definitions (e.g., as in Todorov 2009 or Adamczyk et al. 2023)?
2. Please clarify your RL formulation: what are the exact state and action spaces, and is discounting used? Are the actions restricted to arithmetic tokens or full LLM vocabulary?
3. Can you provide formal definitions and concrete examples for “length generalization” and “compositional generalization,” perhaps illustrated via Fig. 1 patterns?
4. What is the total dataset size and distribution of patterns by shape? How would Fig. 6 (dataset pattern distribution) look?
5. Are results averaged over multiple seeds? If so, please report variance; if not, why?
6. How can the claims in Sections 4.1 and 4.4 be justified without comparison to pretrained and SFT baselines or statistical validation?
7. It seems like the additional reward of 0.1 for correct formatting has a large effect on the pattern coverage performances. How would these results change in the absence of that reward?

---

> ### Author Response · Authors · 2025-11-24
> **Response to Reviewer fN4B (1/4)**
>
> ### Major
>
> > Lack of grounding in RL composition literature: The paper cites none of the foundational and recent works on skill composition in RL, missing a connection to established definitions and frameworks. For example, the foundational work of Todorov [2009] that demonstrated that RL agents could provably compose learned skills to solve new tasks, which was then extended to Boolean composition by Tasse et al [2020] and more general classes of composition operators by Adamczyk et al. [2023ab]. See Mendez et al [2023] for a survey.
>
> We thank the reviewer for the suggestion. In the updated manuscript, we have included discussion of additional related works (e.g., [1]) that discuss skill compositions in LLMs, as well as skill composition in classical RL.
>
> We would like to emphasize that classical RL and RL post-training for LLM reasoning have diverged, so “skill composition” in our setting differs from its use in classical RL, even though they aim to capture similar concepts. The reviewer’s comment naturally reflects the classical view, where a skill is defined at the MDP (or option) level. In contrast, we follow the norm in the LLM literature and define skills at the task level (input–output behavior), since in the LLM setting (where models are typically accessed via an API) there is no standard, mathematically precise notion of composition analogous to the MDP formalization in classical RL.
>
> [1] Chen et al. (2023), “Skill-it! A Data-Driven Skills Framework for Understanding and Training Language Models”
>
> > Ambiguous RL setup: The RL formulation (state, action, discounting, etc.) is unspecified. It is unclear whether the actions correspond to full vocab tokens or restricted arithmetic operators, creating methodological ambiguity.
>
> We thank the reviewer for the comment. Our setup is not formulated as a classical MDP, which is why we did not explicitly specify elements like state, action, or discounting. In line with common practice in RL post-training for LLMs, we treat the model as generating sequences in its native token space, with feedback provided at the sequence level (based on correctness or reward shaping). Thus, the notion of “action” corresponds to token generation (from the full vocab) rather than to a fixed action set such as arithmetic operators. We will clarify this distinction in the paper to avoid confusion with the classical RL formulation.
>
>
> > Non-standard definitions of skills and skill composition: In the paper an atomic skill is defined as action, specifically selecting an operator (-,+,x,/). They then define a skill composition as simply being a regular policy (a sequence of actions corresponding to operators and integers).  However, a skill in RL is usually defined as a policy or value function (that corresponds to some well defined task/MDP) and a skill composition is a set of operators applied to a set of skills to generate a new skill.
>
> As noted above, our use of “skill” follows the LLM/post-training literature rather than the classical, MDP-based RL notion. In our framework, a skill is defined at the task level and we use “atomic skills” as a convenient term for the primitive operations that appear inside those solutions (choosing among the operators -,+,x,/). A “compositional skill” then refers to the ability to chain these primitives into a valid solution program for the task, not to a separate policy/option defined over an MDP. We agree that this terminology can be confusing from a classical RL perspective.
>
>
> > Vague formalism: The main idea in this paper is to disentangle the effect of length and compositional generalization on model generalization. However these two notions of generalization are not mathematically formalized and lack concrete illustrative examples (the definitions given on page 3 are not precise enough). For example Length generalisation is defined as "the ability to apply a known compositional pattern to problems requiring a greater number of atomic operations", but it is unclear what are examples of such problems using the pattern in Figure 1 as the "known" pattern.
>
> We thank the reviewer for pointing this out and have updated our manuscript (see Section 2.2).

---

> ### Author Response · Authors · 2025-11-24
> **Response to Reviewer fN4B (2/4)**
>
> > Missing dataset and statistical details: Dataset size (total number of examples), number of examples per subtree shape, training vs heldout splits, etc are not fully reported. It is also unclear if any of the results are statistically significant, since no measure of variance or statistical significance are provided for the highly stochastic RL results. This is especially problematic for non-critic based RL methods like GRPO that rely on Monte Carlo value estimates.
>
> We would like to note that all dataset details are already reported in Appendix A. We thank the reviewers for their comment about variance, and have updated our manuscript (see Appendix B.1).
>
> > Questionable experimental claims:
> In Section 4.1, the authors claim "Models trained only on n ∈ {3, 4} problems show substantial accuracy improvements on held-out n = 5 puzzles". Improvements over what? Natural baselines like the pretrained model and SFT models are not included for comparison. The authors also claim "These results confirm that models successfully achieve length generalization". This is unjustified given the drastic accuracy decline as n increases. It is also unjustified given that the heldout tasks (n=5,6) look like they are not constrained to only "known" patterns from the training tasks. This is a problem given that the main point of this experiment was to isolate the amount of length generalisation from compositional generalisation. In general this experiment is not properly controlled.
> The claims in Section 4.4 rely solely on the heldout curve in Fig 4 (bottom-left and bottom-center). This experiment is not only not averaged across multiple training seeds, but it is also not averaged across different choices of held-out patterns.
>
> We thank the reviewer for these comments and will clarify the experimental claims. In Section 4.1, “substantial accuracy improvements on held-out (n=5) puzzles” refers to gains over the same pretrained model before RL post-training (and over chance), not over SFT; in the updated manuscript, we have made this more explicit in the text and figure caption.
>
> We also note that two supervised fine-tuning (SFT) baselines on the same COUNTDOWN data are already included in Section 4.5.
>
> Regarding “length generalization,” we use this term to refer to the model’s ability to extend its learned procedure to larger problem sizes than those seen during training. Concretely, we say that the RL fine-tuned model exhibits length generalization if for any unseen length $n>n_{train}$​, it attains higher accuracy than the pretrained model. The decline in accuracy for increasing $n$ simply reflects the increasing difficulty of the task, while the relative improvement for any fixed, unseen $n$ indicates generalization beyond the training regime.
>
> We agree that the length generalization studied in Section 4.1 also entails generalizing to patterns unseen during training (hence “generalization”). However, we treat it as an initial, coarse probe and then use the controlled held-out-pattern setup in Section 4.4 to study compositional generalization more directly.
>
> Finally, for Section 4.4, the main-text curves in Fig. 4 (now Figure 5) show one representative figure, similar to Figure 2 and 3 (now Figures 3 and 4). We had run the same protocol across 3 seeds and observed qualitatively similar held-out curves. These additional results are now included in the Appendix G. Due to limited compute budget, we were not able to run additional experiments on other heldout patterns.
>
>
> ### Minor
>
> > Undefined abbreviations (e.g., SFT on first use)
>
> We thank the reviewer for pointing this out, and have explicitly defined the SFT abbreviation on first use in our updated manuscript.
>
> > The accuracy of the greedy policy (i.e. when selecting the highest probability action) is not reported.
>
> We thank the reviewer for this suggestion. For reasoning tasks (particularly under test-time scaling setups), greedy decoding is not always a standard decoding strategy. Prior works (e.g., [1, 2]) have shown that sampling-based decoding, which explores multiple reasoning paths, is crucial for achieving reliable performance improvements as test-time compute increases. In contrast, greedy decoding collapses this diversity and underestimates a model’s reasoning ability. For this reason, we report results using the standard sampling-based evaluation. Instead, please consider All-correct@32 as a proxy for the Pass@1 accuracy under greedy decoding.
>
> [1] Snell et al. (2024), “Scaling LLM Test-Time Compute Optimally Can be More Effective than Scaling Parameters for Reasoning,” ICLR 2025.
>
> [2] DeepSeek-AI (2025), “DeepSeek-R1: Incentivizing Reasoning Capability in LLMs via Reinforcement Learning,” arXiv preprint.

---

> ### Author Response · Authors · 2025-11-24
> **Response to Reviewer fN4B (3/4)**
>
> > What are those metrics in Fig 2 bottom? We can guess but the paper should precisely define them.
>
>
> We thank the reviewer for pointing this out. We have included explicit pointers for the terms in Section 3.2.
>
> > Table 3 is missing the pretrained model without finetuning.
>
> We thank the reviewer for pointing this out. We have included the missing baseline in our updated manuscript.
>
>
> ### Other
>
> > Overall, this paper makes interesting methodological and empirical contribution to understanding how RL induces compositional reasoning in LLMs, but it would benefit from tighter formal definitions, stronger theoretical grounding, and more controlled experimental validation.
>
> We thank the reviewer for their suggestion, and have updated our manuscript accordingly.
>
> ### Questions
>
> > Why depart from the standard RL definitions of “skills” and “skill composition”? How would the results or interpretations change under conventional definitions (e.g., as in Todorov 2009 or Adamczyk et al. 2023)?
>
> We would first like to clarify that we use **conventional definitions in the domain of language models** (not RL). While LLMs have adopted RL techniques, the two fields have diverged and use different definitions for similar concepts.
>
> Given the ambiguous nature of natural language, the task corresponding to each skill cannot be mathematically formulated as opposed to classical RL setting. For example, how would one mathematically formulate the skill of “using a metaphor”? Instead, in LLM literature, an atomic “skill” generally refers to the competency of the language model on a group of semantically related queries (measured by some skill-specific metric based on the generated output) [1, 2].
>
> In our paper, we mention that “the atomic skills would be mastering the four operators: a+b, a-b, a*b, a/b” (footnote 2 below line 107). This would include the LLM’s ability to 1) identify that a COUNTDOWN puzzle of size 2 can be solved with an equation that involves a specific choice of the operator (procedural skills); 2) correctly evaluate the arithmetic expression (arithmetic skills). The skill composition is defined as the model’s ability to solve puzzles that involve these atomic skills as subroutines. In Section 3.2., we define the metrics to measure the competency of the LLM based on the generated output.
>
> [1] Chen et al. (2023), “Skill-it! A Data-Driven Skills Framework for Understanding and Training Language Models”
>
> [2] Arora et al. (2023), “A Theory for Emergence of Complex Skills in Language Models”
>
>
> > Please clarify your RL formulation: what are the exact state and action spaces, and is discounting used? Are the actions restricted to arithmetic tokens or full LLM vocabulary?
>
> Since we do not depart from the **standard RL post-training procedure for LLMs**, we follow the convention of the LLM literature and do not report such details (e.g., state/action space). While some previous works on COUNTDOWN consider training with a symbolic approach (e.g., using a restricted / special set of tokens during decoding), we explicitly mention that we distinguish our work from such approaches: “no handcrafted search heuristics or **specialized decoding**” (line 443).
>
> To clarify here, we generate a sequence of natural language tokens and assign a reward on each rollout based on the parsed output. The set of actions correspond to the entire vocabulary set. We explicitly mention that “for all other hyperparameters, we use the default values from the verl package version 0.5.0.dev0” (Appendix A.3, line 881, referenced in line 178). This includes the gamma value of 1.0 (no discount) (https://github.com/volcengine/verl/blob/v0.5.x/verl/trainer/config/ppo_trainer.yaml). We also report the prompt template we used in Appendix E (referenced in Appendix A.3, referenced in line 178). This template is a **standard** prompt template for training LLMs with GRPO on reasoning tasks (i.e., asking them to include chain of thought between \<think\>\</think\> tags and to include the final answer between \<answer\>\</answer\> tags) [3].
>
> For more details on the formulation (state / action) of the RL techniques on LLMs, please see more foundational papers (e.g., [4, 5]).
>
> [3] DeepSeek-AI (2025), “DeepSeek-R1: Incentivizing Reasoning Capability in LLMs via Reinforcement Learning,” arXiv preprint.
>
> [4] Ziegler et al. (2019),  “Fine-Tuning Language Models from Human Preferences,” arXiv preprint.
>
> [5] Shao et al. (2024), “DeepSeekMath: Pushing the Limits of Mathematical Reasoning in Open Language Models,” arXiv preprint.

---

> ### Author Response · Authors · 2025-11-24
> **Response to Reviewer fN4B (4/4)**
>
> > Can you provide formal definitions and concrete examples for “length generalization” and “compositional generalization,” perhaps illustrated via Fig. 1 patterns?
>
> As mentioned above, in the LLM domain, it is difficult to make formal definitions of skills. The same is true for concepts like length / compositional generalization.
>
> However, to provide a rough sketch: assume there is a set $\mathcal{S}$ of atomic skills that the model is capable of. Length generalization would be generalizing from elements of $\mathcal{S}^{\leq k_1}$ to elements of $\mathcal{S}^{k_2}$ where $k_1 < k_2$. On the other hand, compositional generalization would be generalizing from elements of $S \subset \mathcal{S}^k$ to elements of $T \subset \mathcal{S}^k$ where $S \cap T = \emptyset$ but the marginal of $T$ (the elements of $\mathcal{S}$ that appear at least once as a coordinate of an element of $T$) is $\mathcal{S}$.
>
> For more details, please see Section 2.2 of the updated manuscript and the response to all reviewers:
> **Clarified Primary Contributions**.
>
> > What is the total dataset size and distribution of patterns by shape? How would Fig. 6 (dataset pattern distribution) look?
>
> This is reported in Appendix A.2.3 (in the same appendix section as Fig. 6. (now Figure 7)): “For each pattern, we [...] have 4000 distinct examples [...]. There are only 15 patterns (4 from n = 3 and 11 from n = 4) where we sample less than 4000 examples. The exact numbers are given in Table 4. For each pattern, we select the first 10 examples as heldout examples. In total, we have 418619 examples for training and 1140 examples for testing.”
>
> > Are results averaged over multiple seeds? If so, please report variance; if not, why?
>
> We thank the reviewer for pointing out the ambiguity in how the results were presented.
>
> Originally, the results in Table 1 and 2 were computed by aggregating the entire results across multiple seeds (i.e., [num1 + num2] / [den1 + den2] instead of ([num1] / [den1] + [num2] + [den2]) / 2). In the updated version of the manuscript, we have updated the numbers to be averaged across multiple seeds. We have also updated the caption to provide more detail.
>
> Note: while updating the tables, we also found a minor indexing error in our analysis code, which resulted in some of the n=3 results to be aggregated with n=4 results. After fixing the error, some numbers for the n=3, n=4 columns have changed in Table 1. However, the main message has not changed.
>
> In the newer version of the manuscript, we have also included a more comprehensive version of Tables 1 and 2 (Appendix B.1) which include the standard deviation. We thank the reviewer for the suggestion.
>
> > How can the claims in Sections 4.1 and 4.4 be justified without comparison to pretrained and SFT baselines or statistical validation?
>
> For the SFT baseline (and the discussion around constructing it), please see Section 4.5.
>
> In the newer version of the manuscript, we have additionally included the numbers for the base model in the more comprehensive version. We thank the reviewer for the suggestion.
>
> > It seems like the additional reward of 0.1 for correct formatting has a large effect on the pattern coverage performances. How would these results change in the absence of that reward?
>
> Could the reviewer clarify what they mean by “has a large effect on the pattern coverage performances”?
>
> In any case, adding a formatting reward is a standard practice for LLM + GRPO on reasoning tasks to accelerate training [1, 2, 3].
>
> [1] DeepSeek-AI (2025), “DeepSeek-R1: Incentivizing Reasoning Capability in LLMs via Reinforcement Learning,” arXiv preprint.
>
> [2] Stojanovski et al. (2025), “REASONING GYM: Reasoning Environments for Reinforcement Learning with Verifiable Rewards,” NeurIPS 2025 Datasets and Benchmarks.
>
> [3] Qwen Team (2025), “Qwen3 Technical Report,” arXiv preprint.

---

### Official Review · Reviewer_6KuU · 2025-11-07

**Soundness:** 3
**Presentation:** 4
**Contribution:** 3
**Rating:** 6
**Confidence:** 3

**Summary:**

This paper digs into how RL actually teaches LLMs to combine skills, using COUNTDOWN (an arithmetic puzzles where you make a target number from given numbers using operators such as +, -, etc.) as a clean testbed. The main contribution is a framework that maps expressions to canonical tree structures, allowing us to track exactly which problem-solving patterns models learn and when. They make an important distinction that prior work missed: length generalization (applying things learnt from shorter sequences to longer sequences) vs. compositional generalization (combining known things in novel ways). Training on n=3,4 problems, models generalize pretty well to n=5,6. Interestingly, it's not "problem length" that determines difficulty, rather it's the compositional structure. They discover "lookahead bottleneck" where right-heavy patterns like A/(B-C/D) are much harder than left-heavy ones at the same depth, because you have to commit to the outer operation before you know what complex thing comes later. The held-out pattern experiments show models can actually synthesize new patterns they've never seen, which is genuine compositional generalization.

**Strengths:**

Clearly separation of two often conflated generalizations: The authors cleanly split length vs. compositional generalization through careful dataset design (making all patterns equally common, which existing datasets don't do). This helps show that RL teaches real skills combination, not just "do the same thing but longer"

The lookahead bottleneck is a cool finding: Right-heavy patterns are fundamentally harder than left-heavy ones even at equal depth. This reveals something about autoregressive models -- they struggle when they need to plan ahead before generating complex sub-parts. That's not about the task, that's about the architecture

Goes way beyond pass@k: They track pattern coverage and precision over training, showing there's a big gap between "the model can generate this pattern sometimes" and "the model reliably executes it." Pass@k completely misses this dynamic, showing a fundamental flaw in the metric.

**Weaknesses:**

The biggest weakness according to me that everything in the paper is just learnt from experiments on COUNTDOWN, which is one synthetic math task. We don't actually know if the lookahead bottleneck shows up in complex tasks such as code generation, theorem proving, or other compositional tasks. The generalizability claims are reasonable speculation but still speculation.

This sort of puts the work in a kind of foundational category (for me), but since the learnings are all from empirical results, it's neither fully experimental (with more elaborate experiments), or foundational (no guarantees as such -- only speculation).

**Questions:**

Quoting the authors “right-heavy structures that demand significant lookahead before executing a complex subroutine constitute a fundamental bottleneck for autoregressive models". What specific feature of the autoregressive models causes this? Are there any fixes that the authors tried?

---

> ### Author Response · Authors · 2025-11-24
> **Response to Reviewer 6KuU**
>
> > The biggest weakness according to me that everything in the paper is just learnt from experiments on COUNTDOWN, which is one synthetic math task. We don't actually know if the lookahead bottleneck shows up in complex tasks such as code generation, theorem proving, or other compositional tasks. The generalizability claims are reasonable speculation but still speculation.
>
> We thank the reviewer for pointing this out. Please see the response to all reviewers **Scope and choice of COUNTDOWN**.
>
> > This sort of puts the work in a kind of foundational category (for me), but since the learnings are all from empirical results, it's neither fully experimental (with more elaborate experiments), or foundational (no guarantees as such -- only speculation).
>
> We appreciate the reviewer’s concern about how to categorize the contribution and will clarify our positioning. Our goal is not to provide general formal guarantees nor to offer a large-scale benchmark, but to introduce a controlled synthetic setting where compositional “skills” can be precisely defined and quantified. On natural data, compositional complexity is confounded with length, making it impossible to isolate true failure modes. A synthetic testbed is therefore methodologically essential to control for these confounders and identify drivers of reasoning failure. Thus, the COUNTDOWN task introduces a formal lens through which to analyze the acquisition of compositional skills.
>
> > Quoting the authors “right-heavy structures that demand significant lookahead before executing a complex subroutine constitute a fundamental bottleneck for autoregressive models". What specific feature of the autoregressive models causes this? Are there any fixes that the authors tried?
>
> We thank the reviewer for raising this point. In our view, the difficulty with right-heavy structures such as A / (B − C / D) arises from the left-to-right next-token prediction in autoregressive models: the model must first commit to “A divided by …” and then correctly maintain this pending operation in working memory while computing the nested subexpression B − C / D, whereas a left-heavy form like (A / B − C) / D can be processed more incrementally with immediate local operations. This mirrors classic observations in cognitive science that right-branching arithmetic expressions place higher demands on working memory, which we hypothesize corresponds to the “lookahead” bottleneck we observe. That said, we do not claim a complete mechanistic account of why expressions like A / (B − C / D) are harder. We also have not yet explored targeted fixes (e.g., alternative decoding orders), and we see systematically probing such interventions as an interesting direction for future work.

---

### Official Review · Reviewer_7q35 · 2025-11-10

**Soundness:** 3
**Presentation:** 4
**Contribution:** 3
**Rating:** 8
**Confidence:** 3

**Summary:**

The paper studies on the synthetic task of Countdown to what degree RL training can induce skill composition. The primary contribution is the design of a specific task in which problems with different solution patterns exhibit different degrees of difficulty, irrespective of the total length. Carefully constructed RL experiments reveal that on the task of Countdown, length generalization is possible, but the order in which certain skills are learned, depends on the type of solution pattern.

**Strengths:**

- the paper is well presented and offers all or many details necessary for reproduction
- the topic is very relevant (RL + LLMs + reasoning) and the paper provides insights into the learning dynamics and limitations which are currently scarce

**Weaknesses:**

- the experimental setup appears to be sound and convincing, albeit it could still be more comprehensive with an additional task besides Countdown
- the SFT baseline is somewhat weak. I understand that having a strong SFT baseline is not the primary goal here, but it would be interesting to study some rejection fine-tuning settings in which one samples from an RLed model correct trajectories and filters them based on the specific distribution one wants to SFT on

**Questions:**

does increasing the proportion of right-heavy training examples also translate to a more balanced generalization gap? is there a correlation between the fraction of non-zero advantage batches for different patterns during training and the generalization performance of that pattern?

---

> ### Author Response · Authors · 2025-11-24
> **Response to Reviewer 7q35**
>
> > the experimental setup appears to be sound and convincing, albeit it could still be more comprehensive with an additional task besides COUNTDOWN
>
> We thank the reviewer for this comment. Please see the response to all reviewers **Scope and choice of COUNTDOWN**
>
> > the SFT baseline is somewhat weak. I understand that having a strong SFT baseline is not the primary goal here, but it would be interesting to study some rejection fine-tuning settings in which one samples from an RLed model correct trajectories and filters them based on the specific distribution one wants to SFT on
>
> We thank the reviewer for this comment. Please first see the response to all reviewers **RL post-training setup and “skills”**. As the reviewer correctly points out, the primary goal of the paper is not to have a strong SFT baseline. Instead, the SFT baseline is presented to show the learning dynamics when not training on RL methods (i.e., training on self-generated outputs).
> We agree that it would also be interesting to see the learning dynamics of other training methods. However, the analysis for rejection finetuning would easily be confounded with the filtering heuristics.
> We hope this would be an interesting direction for a future study.
>
> > does increasing the proportion of right-heavy training examples also translate to a more balanced generalization gap?
>
> We thank the reviewer for the question. We speculate the reviewer’s hypothesis is likely true. In Appendix A.2.1, we identify that existing datasets do not contain enough right-heavy patterns. Then in Appendix A.2.2, we show that training on such a faulty dataset leads to a model that is far worse at right-heavy patterns. Our fix of preparing the same number of examples (thereby increasing the proportion of right-heavy training examples) already improved the generalization gap. We speculate that further increasing the proportion of right-heavy examples would lead to more improvements in the generalization gap.
>
> > is there a correlation between the fraction of non-zero advantage batches for different patterns during training and the generalization performance of that pattern?
>
> We agree that a more fine-grained analysis of the training rewards (not just the evaluation on held-out data) per pattern would be an interesting venue for future works. However, we were not yet able to carry out a careful analysis towards this direction.
> The fraction of non-zero advantage batches would be a metric that would continuously change over the course of training. However, within each batch (i.e., at every gradient update step), there are only 256 examples, so each pattern is expected to appear only around 2 times, so measuring the fraction of such a small number would lead to a high variance in the numbers.

---

### Author Response · Authors · 2025-11-24
**Response to All Reviewers (1/2)**

We thank the reviewers for their thoughtful comments, and for recognizing the novelty of our work. We have uploaded a revised manuscript incorporating additional baselines, statistical variance analysis, and clearer definitions.

Below, we first clarify our primary contributions and then address three themes raised across the reviews:
- (1) the scope and generality beyond COUNTDOWN;
- (2) the clarity of our definitions and RL post-training setup;
- (3) baselines, evaluation protocols, and reproducibility.

### **Clarified Primary Contributions**

1. We refine the distinction between:
- (a) Length generalization: applying a known compositional pattern to puzzles with more atomic operations (often treated in prior work merely as “chaining” skills).
- (b) Compositional generalization: solving held-out patterns at fixed size using the same atomic operations but new tree structures. We provide evidence that RL post-training can induce this form of generalization, rather than just extending the sequence length.
- Section 2.2 now includes revised, explicit definitions and concrete examples.

2. We perform a controlled analysis of RL post-training on compositional structure. We show that problem size is an insufficient proxy for difficulty. For example, a linear pattern like A+B+C+D is significantly easier than a nested pattern like A/(B-C/D), despite identical lengths. Using COUNTDOWN as a testbed, we isolate how RL post-training of LLMs changes the learnability of patterns when length, and in some settings, compositional depth are held constant. We show a clear hierarchy of difficulty (e.g., balanced vs. right-heavy structures) that is not captured by length alone (Sections 4.2-4.4).

3. We reveal simplicity bias and hidden failure modes in existing COUNTDOWN datasets. We show that standard data-generation pipelines oversample easy, addition/multiplication-only patterns and that aggregate metrics such as pass@k mask systematic weaknesses on specific patterns. We propose a simple, reproducible fix (sampling an equal number of puzzles per pattern) which exposes consistent failure modes across model families and sizes (Section 2.3, Appendix A.2).

### **Scope and choice of COUNTDOWN**

We have clarified the scope of our claims. Our goal is _not_ to provide universal guarantees about compositional reasoning in natural language tasks such as code, theorem proving, etc. Instead, we explicitly position COUNTDOWN as a controlled synthetic setting where (i) underlying patterns and atomic operations are precisely defined, and (ii) length and compositional structure can be disentangled without confounders such as ambiguous skill usage or solution length. In other words, a synthetic setting such as COUNTDOWN is necessary because natural language tasks confound compositional complexity with problem size. We now frame extensions to more natural settings (e.g., AIME, FrontierMATH, HLE) as important future work rather than as established conclusions.

---

> ### Author Response · Authors · 2025-11-24
> **Response to All Reviewers (2/2)**
>
> ### **RL post-training setup and “skills”**
>
> Our training setup follows standard RL post-training for LLMs, not classical RL over an explicit environment/MDP.
>
> 1. We treat the model as a sequence generator over its full vocabulary. Rollouts are sampled in natural language (with \<think\> / \<answer\> tags), and scalar rewards are assigned at the sequence level based on correctness plus a small reward for correct formatting.
>
> 2. We clarify that our use of “skills” follows the LLM/post-training literature, where a skill is defined at the task level, rather than as an option/policy over a formal MDP. In COUNTDOWN, the atomic skills are the primitive arithmetic operations as they appear in solutions, and “skill composition” refers to solving patterns that combine these primitives into more complex and deeper tree structures.
>
> 3. We have expanded the related works section to situate our notions relative to prior work on skill composition both in LLMs and in classical RL, while explicitly noting that our formalization is tailored to the RL post-training for LLMs rather than to classical RL training.
>
> 4. We updated the title from “RL” to “RL post-training” to clarify that we study the effect of the RL techniques applied to LLMs and not RL in general.
>
> Finally, we note that the goal of our study is not to evaluate the overall efficacy of RL post-training compared with other post-training methods, but to analyze what kinds of generalization it induces. For completeness, two supervised fine-tuning (SFT) baselines on the same COUNTDOWN data were already included in the original submission (Section 4.5). These SFT baselines do not exhibit the same hierarchy of task learnability observed under RL post-training, suggesting that this hierarchy arises from RL post-training rather than mere exposure to data.
>
> ### **Baselines, evaluation protocols, and reproducibility**
>
> We have strengthened the experimental section as follows:
> 1. Baselines and variance.
> - Tables 1 and 2 now report the best performance of 7B models (rather than final performance) and explicitly average over multiple seeds.
> - We added the missing pretraining model baseline to Table 3.
> - A more complete version of Tables 1-2 with mean and standard deviation across seeds is now included in Appendix B.1.
>
> 2. Decoding and metrics. We clarify our use of sampling based evaluation metrics (pass@k and all-correct@k), in line with recent work on test-time scaling for reasoning tasks, and we avoid making claims specifically about greedy decoding.
>
> 3. Additional runs and plots. We clarify that the main text curves (e.g., in the compositional generalization experiments) show a representative run and provide plots for all successful runs across seeds in Appendix G.
>
> 4. Reproducibility. We provide an anonymized codebase as supplementary material, including: (i) the pattern extraction procedure described in Appendix A.1, (ii) data generation scripts, and (iii) training/evaluation scripts for RL post-training and SFT baselines.
>
> We hope these revisions substantially clarify our contributions, make the RL post-training setup and terminology more transparent, and address the main concerns raised by reviewers.

---

### Meta-Review · Area_Chair_5XuW · 2026-01-02

**Summary:**

This paper investigates how RL post-training induces skill composition in LLMs using the Countdown arithmetic task as a controlled testbed. The main contributions are: (1) a framework distinguishing length generalization from compositional generalization, (2) demonstrating that RL-trained models exhibit a hierarchy of learnability where balanced tree structures are easier than right-heavy ones, and (3) evidence that models can generalize to unseen compositional patterns. Reviews were divided, with scores ranging from 2 to 8.

**Reviewer Concerns:**

- Missing statistical variance analysis (now in Appendix B.1)
- Unclear definitions of skills/composition (Section 2.2 revised with concrete examples)
- Missing baselines (SFT baselines were already present in Section 4.5; pretrained baseline added to Table 3)
- Reproducibility concerns (code released as supplementary material)
- Missing related work on skill composition (added citations)

- Generalizability beyond COUNTDOWN remains the central issue. While authors argue synthetic settings are necessary to disentangle confounders, multiple reviewers remain unconvinced that findings transfer to realistic tasks.
- Reviewer fN4B's concerns about classical RL formalization reflect a paradigm mismatch (authors flagged this); however, the paper could better bridge these communities.
- Reviewer xutx maintained their score citing incremental novelty and presentation clarity despite author responses.

**Reviewer Scores:**

- 7q35 (8): Would likely maintain score; concerns adequately addressed
- 6KuU (6): Would likely maintain; acknowledged contributions but scope concerns remain
- fN4B (2): Might increase slightly (to 3-4) given clarifications, but fundamental concerns about formalism persist
- xutx (2): Explicitly maintained score post-rebuttal
- 1GMS (4): Might increase to 5 given SFT baseline clarification

---

### Decision · Program_Chairs · 2026-01-26

Reject